# DISTRIBUTIONAL GENERALIZATION:
# A NEW KIND OF GENERALIZATION

## ABSTRACT

We introduce a new notion of generalization— Distributional Generalization— which roughly states that outputs of a classifier at train and test time are close *as distributions*, as opposed to close in just their average error. For example, if we mislabel 30% of dogs as cats in the train set of CIFAR-10, then a ResNet trained to interpolation will in fact mislabel roughly 30% of dogs as cats on the *test set* as well, while leaving other classes unaffected. This behavior is not captured by classical generalization, which would only consider the average error and not the distribution of errors over the input domain. Our formal conjectures, which are much more general than this example, characterize the form of distributional generalization that can be expected in terms of problem parameters: model architecture, training procedure, number of samples, and data distribution. We give empirical evidence for these conjectures across a variety of domains in machine learning, including neural networks, kernel machines, and decision trees. Our results thus advance our understanding of interpolating classifiers.

## 1 INTRODUCTION

We begin with an experiment motivating the need for a notion of generalization beyond test error.

**Experiment 1.** *Consider a binary classification version of CIFAR-10, where CIFAR-10 images $x$ have binary labels* Animal/Object. *Take 50K samples from this distribution as a train set, but apply the following label noise: flip the label of cats to* Object *with probability 30%. Now train a WideResNet $f$ to 0 train error on this train set. How does the trained classifier behave on test samples? Options below:*

1. The test error is low across all classes, since there is only 3% label noise in the train set

2. Test error is "spread" across the animal class, After all, the classifier is not explicitly told what a cat or a dog is, just that they are all animals.

3. The classifier misclassifies roughly 30% of test cats as "objects", but all other types of animals are largely unaffected.

In fact, reality is closest to option (3). Figure 1 shows the results of this experiment with a WideResNet. The left panel shows the joint density of train inputs $x$ with train labels Object/Animal. Since the classifier is interpolating, the classifier outputs on the train set are identical to the left panel. The right panel shows the *classifier predictions $f(x)$* on *test inputs $x$*.

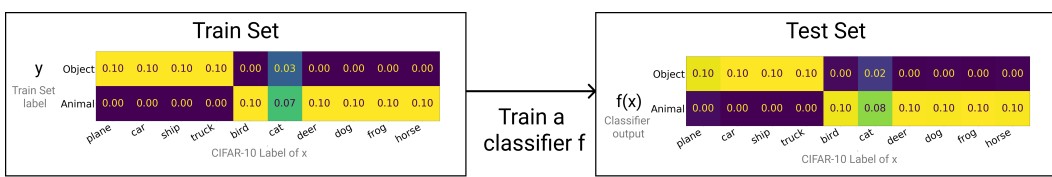

Figure 1: The setup and result of Experiment 1. The CIFAR-10 train set is labeled as either Animals or Objects, with label noise affecting only cats. A WideResNet-28-10 is then trained to 0 train error on this train set, and evaluated on the test set. Full experimental details are in C.2

There are several notable things about this experiment. First, the error is *localized* to cats in test set as it was in the train set, even though no explicit cat labels were provided. Second, the *amount* of error on the cat class is close to the noise applied on the train set. Thus, the behavior of the classifier on the train set *generalizes* to the test set in a certain sense. This type of similarity in behavior would not be captured solely by average test error — it requires reasoning about the entire distribution of classifier outputs. In our work, we show that this experiment is just one instance of a *different type of generalization*, which we call "Distributional Generalization". We now describe the mathematical form of this generalization. Then, through extensive experiments, we will show that this type of generalization occurs widely in existing machine learning methods: neural networks, kernel machines and decision trees.

## 1.1 DISTRIBUTIONAL GENERALIZATION

Supervised learning aims to learn a model that correctly classifies inputs $x \in \mathcal{X}$ from a given distribution $\mathcal{D}$ into classes $y \in \mathcal{Y}$. We want a model with small *test error* on this distribution. In practice, we find such a classifier by minimizing the *train error* of a model on the train set. This procedure is justified when we expect a small *generalization gap*: the gap between the error on the train and test set. That is, the trained model $f$ should have: $\mathrm{Error}_{\mathrm{TrainSet}}(f) \approx \mathrm{Error}_{\mathrm{TestSet}}(f)$. We now re-write this classical notion of generalization in a form better suited for our extension.

**Classical Generalization:** *Let $f$ be a trained classifier. Then $f$ generalizes if:*

$$\mathop{\mathbb{E}}_{\substack{x \sim \mathit{TrainSet} \\ \widehat{y} \leftarrow f(x)}} [\mathbb{1}\{\widehat{y} \neq y(x)\}] \approx \mathop{\mathbb{E}}_{\substack{x \sim \mathit{TestSet} \\ \widehat{y} \leftarrow f(x)}} [\mathbb{1}\{\widehat{y} \neq y(x)\}] \tag{1}$$

Above, $y(x)$ is the true class of $x$ and $\widehat{y}$ is the predicted class. The LHS of Equation 1 is the train error of $f$, and the RHS is the test error. Crucially, both sides of Equation 1 are expectations of the same function ($T_{\mathrm{err}}(x, \widehat{y}) := \mathbb{1}\{\widehat{y} \neq y(x)\}$) under different distributions. The LHS of Equation 1 is the expectation of $T_{\mathrm{err}}$ under the "Train Distribution" $\mathcal{D}_{\mathrm{tr}}$, which is the distribution over $(x, \widehat{y})$ given by sampling a train point $x$ along with its classifier-label $f(x)$. Similarly, the RHS is under the "Test Distribution" $\mathcal{D}_{\mathrm{te}}$, which is this same construction over the test set. These two distributions are the central objects in our study, and are defined formally in Section 2.1. We can now introduce Distributional Generalization, which is a property of trained classifiers. It is parameterized by a set of bounded functions ("tests"): $\mathcal{T} \subseteq \{T : \mathcal{X} \times \mathcal{Y} \rightarrow [0, 1]\}$.

**Distributional Generalization:** *Let $f$ be a trained classifier. Then $f$ satisfies Distributional Generalization with respect to tests $\mathcal{T}$ if:*

$$\forall T \in \mathcal{T} : \quad \mathop{\mathbb{E}}_{\substack{x \sim \mathit{TrainSet} \\ \widehat{y} \leftarrow f(x)}} [T(x, \widehat{y})] \approx \mathop{\mathbb{E}}_{\substack{x \sim \mathit{TestSet} \\ \widehat{y} \leftarrow f(x)}} [T(x, \widehat{y})] \tag{2}$$

We write this property as $\mathcal{D}_{\mathrm{tr}} \approx^{\mathcal{T}} \mathcal{D}_{\mathrm{te}}$. This states that the train and test distribution have similar expectations for all functions in the family $\mathcal{T}$. For the singleton set $\mathcal{T} = \{T_{\mathrm{err}}\}$, this is equivalent to classical generalization, but it may hold for much larger sets $\mathcal{T}$. For example in Experiment 1, the train and test distributions match with respect to the test function "*Fraction of true cats labeled as object*." In fact, it is best to think of Distributional Generalization as stating that the distributions $\mathcal{D}_{\mathrm{tr}}$ and $\mathcal{D}_{\mathrm{te}}$ are close *as distributions*.

This property becomes especially interesting for *interpolating classifiers*, which fit their train sets exactly. Here, the Train Distribution $(x_i, f(x_i))$ is exactly equal[1] to the original distribution $(x, y) \sim \mathcal{D}$, since $f(x_i) = y_i$ on the train set. In this case, distributional generalization claims that the output distribution $(x, f(x))$ of the model on test samples is close to the *true* distribution $(x, y)$. The following conjecture specializes Distributional Generalization to interpolating classifiers, and will be the main focus of our work.

**Interpolating Indistinguishability Meta-Conjecture (informal):** *For interpolating classifiers $f$, and a large family $\mathcal{T}$ of test functions, the distributions:*

$$\boxed{(x, f(x))_{x \in \mathit{TestSet}} \ \approx^{\mathcal{T}} \ (x, f(x))_{x \in \mathit{TrainSet}} \ \equiv \ (x, y)_{x, y \sim \mathcal{D}}}$$

---

[1] The formal definition of Train Distribution, in Section 2.1, includes the randomness of sampling the train set as well. We consider a fixed train set in the Introduction for sake of exposition.

This is a "meta-conjecture", which becomes a concrete conjecture once the family of tests $\mathcal{T}$ is specified. One of the main contributions of our work is formally stating two concrete instances of this conjecture— specifying exactly the family of tests $\mathcal{T}$ and their dependence on problem parameters (the distribution, model family, training procedure, etc). It captures behaviors far more general than Experiment 1, and applies to neural networks, kernels, and decision trees. We give empirical evidence for conjectures across a variety of natural settings in machine learning.

## 1.2 SUMMARY OF CONTRIBUTIONS

We extend the classical framework of generalization by introducing Distributional Generalization, in which the train and test behavior of models are close *as distributions*. Informally, for trained classifiers $f$, its outputs on the train set $(x, f(x))_{x \in \text{TrainSet}}$ are close in distribution to its outputs on the test set $(x, f(x))_{x \in \text{TestSet}}$, where the form of this closeness depends on specifics of the model, training procedure, and distribution. This notion is more fine-grained than classical generalization, since it considers the entire distribution of model outputs instead of just the test error.

We initiate the study of Distributional Generalization across various domains in machine learning. For interpolating classifiers, we state two formal conjectures which predict the form of distributional closeness that can be expected for a given model and task:

1. Feature Calibration Conjecture (Section 3): Interpolating classifiers, when trained on samples from a distribution, will match this distribution up to all "distinguishable features" (Definition 1).
2. Agreement Conjecture (Section 4): For two interpolating classifiers of the same type, trained independently on the same distribution, their *agreement probability* with each other on test samples roughly matches their *test accuracy*.

We perform a number of experiments surrounding these conjectures, which reveal new behaviors of standard interpolating classifiers (e.g. ResNets, MLPs, kernels, decision trees). We prove our conjectures for 1-Nearest-Neighbors (Theorem 1), which suggests some form of "locality" as the underlying mechanism. Finally, we discuss extending these results to non-interpolating methods in Section 5. Our experiments and conjectures shed new light on the structure of interpolating classifiers, which are extensively studied in recent years yet still poorly understood.

**Related Work.** Our work is inspired by the broader study of interpolating and overparameterized methods in machine learning (e.g. Zhang et al. (2016); Belkin et al. (2018a;b; 2019); Liang and Rakhlin (2018); Nakkiran et al. (2020); Schapire et al. (1998); Breiman (1995)). In a similar vein to our work, Wyner et al. (2017); Olson and Wyner (2018) investigate decision trees, and show that random forests are equivalent to a Nadaraya–Watson smoother (Nadaraya, 1964; Watson, 1964) with a certain smoothing kernel. Our conjectures also describe neural networks under label noise, which has been empirically and theoretically studied in the past (Zhang et al., 2016; Belkin et al., 2018b; Rolnick et al., 2017; Natarajan et al., 2013; Thulasidasan et al., 2019; Ziyin et al., 2020; Chatterji and Long, 2020), though not formally characterized. The behaviors we consider are also similar to conditional density estimation (e.g. Tsybakov (2008); Dutordoir et al. (2018)), though we consider *samplers*, not density estimators. We include a full discussion of related works in Appendix A.

## 2 PRELIMINARIES

**Notation.** We consider joint distributions $\mathcal{D}$ on $x \in \mathcal{X}$ and discrete $y \in \mathcal{Y} = [k]$. Let $\mathcal{D}^n$ denote $n$ iid samples from $\mathcal{D}$ and $S = \{(x_i, y_i)\}$ denote a train set. Let $\mathcal{F}$ denote the classifier family (including architecture and training algorithm for neural networks), and let $f \leftarrow \text{Train}_{\mathcal{F}}(S)$ denote training a classifier $f \in \mathcal{F}$ on train-set $S$. We consider classifiers which output hard decisions $f : \mathcal{X} \to \mathcal{Y}$. Let $\text{NN}_S(x) = x_i$ denote the nearest-neighbor to $x$ in train-set $S$, with respect to a distance metric $d$. Our theorems will apply to any distance metric, and so we leave this unspecified. Let $\text{NN}_S^{(y)}(x)$ denote the nearest-neighbor estimator itself, that is, $\text{NN}_S^{(y)}(x) := y_i$ where $x_i = \text{NN}_S(x)$.

**Experimental Setup.** Full experimental details are provided in Appendix B. Briefly, we train all classifiers to interpolation (to 0 train error). Neural networks (MLPs and ResNets (He et al., 2016))

are trained with SGD. Interpolating decision trees are trained using the growth rule from Random Forests (Breiman, 2001). For kernel classification, we consider kernel regression on one-hot labels and kernel SVM, with small or 0 of regularization (which is often optimal Shankar et al. (2020)).

**Distributional Closeness.** We consider the following notion of closeness for two probability distributions: For two distributions $P, Q$ over $\mathcal{X} \times \mathcal{Y}$, let a "test" (or "distinguisher") be a function $T : \mathcal{X} \times \mathcal{Y} \to [0, 1]$ which accepts a sample from either distribution, and is intended to classify the sample as either from distribution $P$ or $Q$. For any set $\mathcal{C} \subseteq \{T : \mathcal{X} \times \mathcal{Y} \to [0, 1]\}$ of tests, we say distributions $P$ and $Q$ are "$\varepsilon$-indistinguishable up to $\mathcal{C}$-tests" if they are close with respect to all tests in class $\mathcal{C}$. That is,

$$P \approx_\varepsilon^\mathcal{C} Q \iff \sup_{T \in \mathcal{C}} \left| \mathop{\mathbb{E}}_{(x,y) \sim P} [T(x, y)] - \mathop{\mathbb{E}}_{(x,y) \sim Q} [T(x, y)] \right| \leq \varepsilon \qquad (3)$$

Total-Variation distance is equivalent to closeness in all tests, i.e. $\mathcal{C} = \{T : \mathcal{X} \times \mathcal{Y} \to [0, 1]\}$, but we consider closeness for restricted families of tests $\mathcal{C}$. $P \approx_\varepsilon Q$ denotes $\varepsilon$-closeness in TV-distance.

## 2.1 Framework for Indistinguishability

We consider throughout the following three distributions over $\mathcal{X} \times \mathcal{Y}$:

| **Source $\mathcal{D}$:** $(x, y)$ where $x, y \sim \mathcal{D}$ | **Train $\mathcal{D}_{tr}$:** $(x_{tr}, f(x_{tr}))$ $S \sim \mathcal{D}^n, f \leftarrow \text{Train}_\mathcal{F}(S),$ $x_{tr}, y_{tr} \sim S$ | **Test $\mathcal{D}_{te}$** $(x, f(x))$ $S \sim \mathcal{D}^n, f \leftarrow \text{Train}_\mathcal{F}(S),$ $x, y \sim \mathcal{D}$ |
| --- | --- | --- |

The **Source Distribution** $\mathcal{D}$ is simply the original distribution. To sample from the **Train Distribution $\mathcal{D}_{tr}$**, we first sample a train set $S \sim \mathcal{D}^n$, train a classifier $f$ on it, then output $(x_{tr}, f(x_{tr}))$ for a random *train point* $x_{tr}$. That is, $\mathcal{D}_{tr}$ is the distribution of input and outputs of a trained classifier $f$ on its train set. To sample from the **Test Distribution $\mathcal{D}_{te}$**, do we this same procedure, but output $(x, f(x))$ for a random *test point* $x$. That is, the $\mathcal{D}_{te}$ is the distribution of input and outputs of a trained classifier $f$ at test time. The only difference between the Train Distribution and Test Distribution is that the point $x$ is sampled from the train set or the test set, respectively.[2] For interpolating classifiers, $f(x_{tr}) = y_{tr}$ on the train set, and so the Source and Train distributions are equivalent: $\mathcal{D} \equiv \mathcal{D}_{tr}$. Our general thesis is that the Train and Test Distributions are indistinguishable under a variety of test families $\mathcal{T}$. Formally, we argue that for certain families of tests $\mathcal{T}$ and interpolating classifiers $\mathcal{F}$, the distributions: $\mathcal{D} \equiv \mathcal{D}_{tr} \approx_\varepsilon^\mathcal{T} \mathcal{D}_{te}$. Sections 3 and 4 give specific families of tests $\mathcal{T}$ for which these distributions are indistinguishable.

## 3 Feature Calibration

The distributional closeness of Experiments 1 is subtle, and depends on the classifier architecture, distribution, and training method. For example, Experiment 1 does not hold if we use a fully-connected network (MLP) instead of a ResNet, or if we early-stop the ResNet instead of training to interpolation (see Appendix C.2). Both these scenarios fail in different ways: An MLP cannot properly distinguish cats even when trained on real CIFAR-10 labels, and so (informally) it has no hope of behaving differently on cats in the setting of Experiment 1. On the other hand, an early-stopped ResNet for Experiment 1 does not label 30% of cats as objects on the *train set*, since it does not interpolate, and thus has no hope of reproducing this behavior on the test set. We now characterize these behaviors, and their dependency on problem parameters, via a formal conjecture.

This conjecture characterizes a family of tests $\mathcal{T}$ for which the output distribution of a classifier $(x, f(x)) \sim \mathcal{D}_{te}$ is "close" to the source distribution $(x, y) \sim \mathcal{D}$. At a high level, we argue that the distributions $\mathcal{D}_{te}$ and $\mathcal{D}$ are statistically close if we first "coarsen" the domain of $x$ by some labelling $L : \mathcal{X} \to [M]$. That is, for certain partitions $L$, the following distributions are statistically close: $(L(x), f(x)) \approx_\varepsilon (L(x), y)$. Intuitively, allowable partitions are those which can be learnt from samples. To formalize the set of allowable partitions $L$ for a training algorithm, we define a

---

[2]Technically, these definitions require training a fresh classifier for each sample, using independent train sets. We use this definition because we believe it is natural, although for practical reasons most of our experiments train a single classifier $f$ and evaluate it on the entire train/test set.

*distinguishable feature*: a partition of the domain $\mathcal{X}$ that is learnable for a given family of models. For example, in Experiment 1, the partition into CIFAR-10 classes would be a distinguishable feature for ResNets, but not for MLPs.

**Definition 1** (($\varepsilon, \mathcal{F}, \mathcal{D}, n$)-Distinguishable Feature). *For a distribution $\mathcal{D}$ over $\mathcal{X} \times \mathcal{Y}$, number of samples $n$, family of models $\mathcal{F}$, and small $\varepsilon \geq 0$, an $(\varepsilon, \mathcal{F}, \mathcal{D}, n)$-distinguishable feature is a partition $L : \mathcal{X} \to [M]$ of the domain $\mathcal{X}$ into $M$ parts, such that training a model from $\mathcal{F}$ on $n$ samples labeled by $L$ works to classify $L$ with high test accuracy. Precisely, $L$ is a $(\varepsilon, \mathcal{F}, \mathcal{D}, n)$-distinguishable feature if:*

$$\Pr_{\substack{S=\{(x_i, L(x_i)\}_{x_1, \ldots, x_n \sim \mathcal{D}} \\ f \leftarrow \mathrm{Train}_{\mathcal{F}}(S); \ x \sim \mathcal{D}}} [f(x) = L(x)] \geq 1 - \varepsilon$$

This definition depends only on the marginal distribution of $\mathcal{D}$ on $x$, and not on the label distribution $p_{\mathcal{D}}(y|x)$. To recap, this definition is meant to capture a labeling of the domain $\mathcal{X}$ that is learnable for a given training procedure. It must depend on the classifier family $\mathcal{F}$ and number of samples $n$, since more powerful classifiers can distinguish more features. Note that there could be many distinguishable features for a given setting $(\varepsilon, \mathcal{F}, \mathcal{D}, n)$ — including features not implied by the class label such as the presence of grass in a CIFAR-10 image. Our main conjecture in this section is that the test distribution $(x, f(x)) \sim \mathcal{D}_{\mathrm{te}}$ is statistically close to the source distribution $(x, y) \sim \mathcal{D}$ when the domain is "coarsened" by a distinguishable feature. Formally:

**Conjecture 1** (Feature Calibration). *For all natural distributions $\mathcal{D}$, number of samples $n$, family of interpolating models $\mathcal{F}$, and $\varepsilon \geq 0$, the following distributions are statistically close for all $(\varepsilon, \mathcal{F}, \mathcal{D}, n)$-distinguishable features $L$:*

$$\underset{f \leftarrow \mathrm{Train}_{\mathcal{F}}(\mathcal{D}^n); \ x, y \sim \mathcal{D}}{(L(x), f(x))} \quad \approx_{\varepsilon} \quad \underset{x, y \sim \mathcal{D}}{(L(x), y)} \tag{4}$$

We claim that this holds *for all* distinguishable features $L$ "automatically" – we simply train a classifier, without specifying any particular partition. As a trivial instance of the conjecture, suppose we have a distribution with deterministic labels, and consider the $\varepsilon$-distinguishable feature $L(x) := y(x)$, i.e. the label itself. The $\varepsilon$ here is then simply the test error of $f$, and Conjecture 1 is true by definition. The formal statements of Definition 1 and Conjecture 1 may seem somewhat arbitrary, involving many quantifiers over $(\varepsilon, \mathcal{F}, \mathcal{D}, n)$. However, we believe these statements are natural. To support this, in Theorem 1 we prove that Conjecture 1 is formally true as stated for 1-Nearest-Neighbor classifiers.

**Connection to Indistinguishability.** Conjecture 1 is an instantiation of our general Indistinguishably Conjecture. In particular, Conjecture 1 is equivalent to the statement $\mathcal{D}_{\mathrm{te}} \approx_{\varepsilon}^{\mathcal{L}} \mathcal{D}$ where $\mathcal{L}$ is the family of all tests which depend on $x$ only via a distinguishable feature $L$. In other words, $\mathcal{D}_{\mathrm{te}}$ is indistinguishable from $\mathcal{D}$ to any function that only sees the input $x$ via a distinguishable feature $L(x)$.

## 3.1 Experiments

We now give empirical evidence for our conjectures in a variety of settings in machine learning, including neural networks, kernel machines, and decision trees. Selected experiments are summarized here, with full details and further experiments in Appendix C.

**Constant Partition:** Consider the trivially-distinguishable *constant* feature $L(x) = 0$. Then, Conjecture 1 states that the marginal distribution of class labels for any interpolating classifier $f(x)$ is close to the true marginals $p(y)$. We construct a variant of CIFAR-10 with class-imbalance and train classifiers with varying levels of test errors (9-41%) to interpolation on it. The marginals of the classifier outputs are close to that of the train set, irrespective of the test error. (See Appendix C.3).

**Class Partition:** We now consider settings where the original class labels form a distinguishable feature (eg: CIFAR-10 classes are distinguishable by ResNets). The conjecture holds for arbitrary distributions $p(y|x)$. This includes the setting of Experiments 1 from the Introduction. We mislabel class $0 \to 1$ with probability $p$ in the CIFAR-10 train set and train a WideResNet-28-10 (WRN-28-10) on this distribution. Now, we observe the fraction of samples mislabeled by this network $\widehat{p}$ from $0 \to 1$ in the test set. Figure 2 shows $p$ versus $\widehat{p}$. The Bayes optimal classifier for this distribution behaves as a step function (in red), and a classifier that obeys Conjecture 1 exactly would follow the diagonal (in green). The actual experiment (in blue) is close to the behavior predicted by Conjecture 1.

Appendix C.4 includes experiments for more distributions (including random joint density matrices) and other classifiers such as Decisions Trees.

**Multiple features:** Conjecture 1 states that the network should be automatically calibrated for all distinguishable features, without any explicit labels for them. To verify this, we use the CelebA dataset (Liu et al., 2015), containing images with many binary attributes per image. We train a ResNet-50 to classify one of the hard attributes (accuracy 80%) and confirm that the output distribution is calibrated with respect to all attributes (Figure 2 that are themselves distinguishable by a ResNet-50.

**Coarse Partition:** Consider AlexNet trained on ILSVRC-2012 ImageNet (Russakovsky et al., 2015), a 1000-class image classification problem with 116 varieties of dogs. The network achieves only 56.5% accuracy on the test set. But it will at least classify most dogs as dogs (with 98.4% accuracy), making $L(x) \in \{\text{dog, not-dog}\}$ a distinguishable feature. Per Conjecture 1, the network is *calibrated* with respect to dogs: 22.4% of all dogs in ImageNet are Terriers, and the network classifies 20.9% of all dogs as Terriers (though it has 9% error on which specific dogs it classifies as Terriers). See Appendix Table 2 for details, and related experiments on ResNets and kernels in Appendix C.

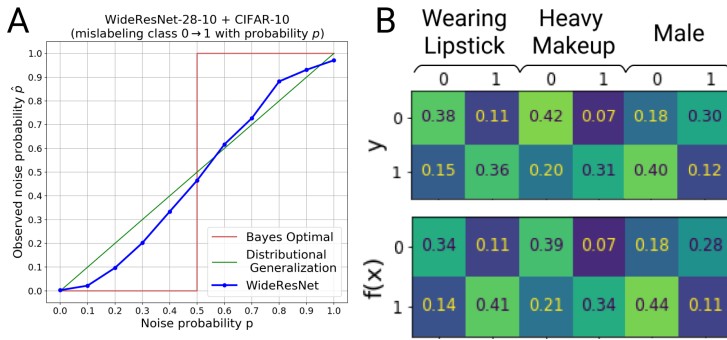

Figure 2: **Feature Calibration.** (A) CIFAR-10 with $p$ fraction of class $0 \to 1$ mislabeled. Actual $p$ vs. observed noise in the classifier outputs. (B) Multiple feature calibration on CelebA.

## 3.2 DISCUSSION

Conjecture 1 claims that $\mathcal{D}_{\text{te}}$ is close to $\mathcal{D}$ up to all tests which are *themselves learnable*. That is, if an interpolating method is capable of learning a certain partition of the domain, then it will also produce outputs that are calibrated with respect to this partition, when trained on any problem. This conjecture thus gives a way of quantifying the resolution with which classifiers approximate the source distribution $\mathcal{D}$, via properties of the classification algorithm itself. This is in contrast to many classical ways of quantifying the approximation of density estimators, which rely on *analytic* (rather than *operational*) distributional assumptions (Tsybakov, 2008; Wasserman, 2006).

**Proper Scoring Rules.** If the loss function used in training is a *strictly-proper scoring rule* such as cross-entropy (Gneiting and Raftery, 2007), then we may expect that in the limit of a large-capacity network and infinite data, training on samples $\{(x_i, y_i)\}$ will yield a good density estimate of $p(y|x)$ at the softmax layer. However, this is not what is happening in our experiments: First, our experiments consider the hard-decisions, not the softmax outputs. Second, we observe Conjecture 1 even in settings without proper scoring rules (e.g. kernel SVM and decision trees).

**1-Nearest-Neighbors Connection.** Here we show that the 1-Nearest-Neighbor classifier provably satisfies Conjecture 1, under mild assumptions. This theorem applies generically to a wide class of distributions, with no assumptions on the ambient dimension of inputs or the underlying metric. The only assumption is a weak regularity condition: sampling the nearest-neighbor train point to a random test point should yield (close to) a uniformly random test point.

**Theorem 1.** *Let $\mathcal{D}$ be a distribution over $\mathcal{X} \times \mathcal{Y}$, and let $n \in \mathbb{N}$ be the number of train samples. Assume the following regularity condition holds: Sampling the nearest-neighbor train point to a random test point yields (close to) a uniformly random test point. That is, suppose that for some small $\delta \geq 0$, the distributions:* $\{\text{NN}_S(x)\}_{\substack{S \sim \mathcal{D}^n \\ x \sim \mathcal{D}}} \approx_\delta \{x\}_{x \sim \mathcal{D}}$. *Then, Conjecture 1 holds. That is, for all $(\varepsilon, \text{NN}, \mathcal{D}, n)$-distinguishable partitions $L$, the following distributions are statistically close:*

$$\{(y, L(x))\}_{x,y \sim \mathcal{D}} \quad \approx_{\varepsilon + \delta} \quad \{(\text{NN}_S^{(y)}(x), L(x))\}_{\substack{S \sim \mathcal{D}^n \\ x,y \sim \mathcal{D}}} \tag{5}$$

The proof of Theorem 1 is straightforward, and provided in Appendix E. We view this theorem both as support for our formalism of Conjecture 1, and as evidence that the classifiers we consider in this work have *local* properties similar to 1-Nearest-Neighbors.

## 4 AGREEMENT PROPERTY

We now present an "agreement property" of interpolating classifiers. This property is independent of the previous section, though both are special cases of our general indistinguishability conjecture. We claim that, informally, the test accuracy of a classifier is close to the probability that it agrees with an identically-trained classifier on a disjoint train set.

**Conjecture 2** (Agreement Property). *For certain classifier families $\mathcal{F}$ and distributions $\mathcal{D}$, the test accuracy of a classifier is close to its* agreement probability *with an independently-trained classifier. That is, let $S_1, S_2$ be disjoint train sets sampled from $\mathcal{D}^n$, and let $f_1, f_2$ be classifiers trained on $S_1, S_2$ respectively, then*

$$\Pr_{\substack{f_1 \\ (x,y)\sim\mathcal{D}}} [f_1(x) = y] \approx \Pr_{\substack{f_1, f_2 \\ (x,y)\sim\mathcal{D}}} [f_1(x) = f_2(x)] \tag{6}$$

*Moreover, this holds with high probability over training $f_1, f_2$:* $\Pr_{(x,y)\sim\mathcal{D}}[f_1(x) = y] \approx \Pr_{(x,y)\sim\mathcal{D}}[f_1(x) = f_2(x)]$.

The agreement property may be surprising for several reasons. For example, suppose we have two classifiers $f_1, f_2$ which were trained on independent train sets, and both achieve test accuracy say 50% on a 10-class problem. Depending on our intuition, we may expect (1) They agree with each other much less than they agree with the true label, since each individual classifier is an independently noisy version of the truth. OR (2) They agree with each other much more than 50%, since classifiers tend to have "correlated" predictions. However, in practice there is a surprising coincidence, and they agree with each other very close to 50%. Conjecture 2 also provably holds for 1-Nearest-Neighbors in some settings, under stronger assumptions (Theorem 2 in Appendix E). Finally, in Section D.3, we consider, and refute, several potential mechanisms which could explain the experimental results of Conjecture 2.

**Connection to Indistinguishability.** Conjecture 2 is in fact a special case of our general Indistinguishability Conjecture. Formally, consider the specific test $T_{\text{agree}} : (x, \widehat{y}) \mapsto \mathbb{1}\{f_1(x) = \widehat{y}\}$ where $f_1 \leftarrow \text{Train}_{\mathcal{F}}(\mathcal{D}^n)$. The expectation of this test under the Source Distribution $\mathcal{D}$ is exactly the LHS of Equation 6, while the expectation under the Test Distribution $\mathcal{D}_{\text{te}}$ is exactly the RHS. Thus, Conjecture 2 can be equivalently stated as $\mathcal{D} \approx^{T_{\text{agree}}} \mathcal{D}_{\text{te}}$.

### 4.1 EXPERIMENTS

In our experiments, we train a pair of classifiers $f_1, f_2$ on random disjoint subsets of the train set for a given distribution. Both classifiers are otherwise trained identically, using the same architecture, number of train-samples $n$, and optimizer. We then plot the test error of $f_1$ against the agreement probability $\Pr_x[f_1(x) = f_2(x)]$. Figure 3 shows experiments with ResNet18 on CIFAR-10 and CIFAR-100, with varying train samples $n$. The Agreement Property approximately holds for all pairs of iden-

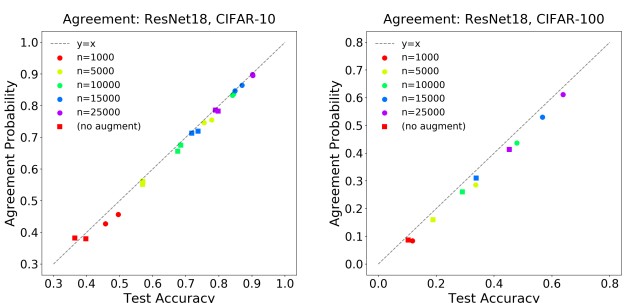

Figure 3: **Agreement Property on CIFAR-10/100.**

tical classifiers, and continues to hold even for "weak" classifiers (e.g. when $f_1, f_2$ have high test error). Full experimental details are in Appendix D, including further experiments with RBF and Laplace kernels, the Myrtle10 kernel (Shankar et al., 2020), and decision trees on UCI tasks.

## 5 Distributional Generalization: Beyond Interpolating Methods

The previous sections have focused on *interpolating* classifiers, which fit their train sets exactly. For non-interpolating classifiers, their outputs on the train set $(x, f(x))_{x \sim \text{TrainSet}}$ will *not* match the original distribution $(x, y) \sim \mathcal{D}$. Thus, there is little hope that their outputs on the test set will match the original distribution, and we do not expect the Indistinguishability Conjecture to hold. However, Distributional Generalization does not require interpolation, and we could still expect that the the train and test distributions are close ($\mathcal{D}_{\text{tr}} \approx^{\mathcal{T}} \mathcal{D}_{\text{te}}$) for some family of tests $\mathcal{T}$. For example, the following is a possible generalization of Feature Calibration.

**Conjecture 3** (Generalized Feature Calibration, informal). *For trained classifiers $f$, the following distributions are statistically close for many partitions $L$ of the domain:*

$$\underset{x_i \sim \text{TrainSet}}{(L(x_i), f(x_i))} \quad \approx \quad \underset{x \sim \text{TestSet}}{(L(x), f(x))} \tag{7}$$

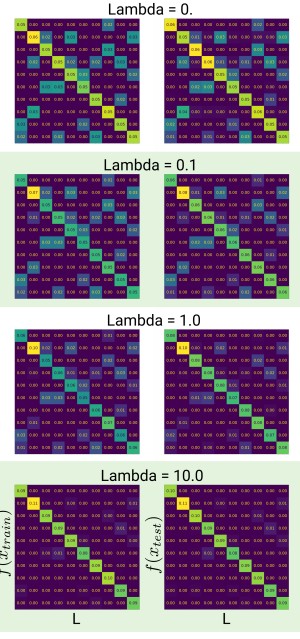

We leave unspecified the exact set of partitions $L$ for which this holds, since we do not yet understand the appropriate notion of "distinguishable feature" in this setting. However, we give experimental evidence suggesting some refinement of Conjecture 3 is true. In Figure 4 we train Gaussian kernel regression on MNIST, with label noise determined by a random sparse confusion matrix. We vary the $\ell_2$ regularization, and plot the confusion matrix of predictions on the train and test sets. With higher regularization, the kernel no longer interpolates the train set, but the test and train confusion matrices remain close. That is, regularization prevents the kernel from fitting the noise on both the train and test sets in a similar way. Full experimental details are given in Appendix B, including an analogous experiment for neural networks on CIFAR-10, with early-stopping in place of regularization (Figure 22). These experiments suggests that Distributional Generalization is a meaningful notion even for non-interpolating classifiers.

Figure 4: **Distributional Generalization**

## 6 Conclusion and Discussion

In this work, we presented a new set of empirical behaviors of standard interpolating classifiers. We unified these under the framework of Distributional Generalization, which states that outputs of trained classifiers on the test set are "close" in distribution to their outputs on the train set. For interpolating classifiers, we stated several formal conjectures (Conjectures 1 and 2) to characterize the form of distributional closeness that can be expected.

**Beyond Test Error.** Our work proposes studying the *entire distribution* of classifier outputs on test samples, beyond just its test error. We show that this distribution is often highly structured, and we take steps towards characterizing it. Surprisingly, modern interpolating classifiers appear to satisfy certain forms of distributional generalization "automatically," despite being trained to simply minimize train error. This even holds in cases when satisfying distributional generalization is in conflict with satisfying classical generalization— that is, when a distributionally-generalizing classifier must necessarily have high test error (e.g. Experiments 1 and 2). We thus hope that studying distributional generalization will be useful to better understand modern classifiers, and to understand generalization more broadly.

**Classical Generalization.** Our framework of Distributional Generalization can be insightful even to study classical generalization. That is, even if we ultimately want to understand test error, it may be easier to do so through distributional generalization. This is especially relevant for understanding the success of interpolating methods, which pose challenges to classical theories of generalization. Our work shows new empirical behaviors of interpolating classifiers, as well as conjectures characterizing these behaviors. This sheds new light on these poorly understood methods, and could pave the way to better understanding their generalization.

**Interpolating vs. Non-interpolating Methods.** Our work also suggests that interpolating classifiers should be viewed as conceptually different objects from non-interpolating ones, even if both have the same test error. In particular, an interpolating classifier will match certain aspects of the original distribution, which a non-interpolating classifier will not. This also suggests, informally, that interpolating methods should not be seen as methods which simply "memorize" their training data in a naive way (as in a look up table) – rather this "memorization" strongly influences the classifier's decision boundary (as in 1-Nearest-Neighbors).

**Limitations.** The conjectures presented in this work are not fully specified, since they do not exactly specify which classifiers or distributions for which they hold. We experimentally demonstrate instances of these conjectures in various "natural" settings in machine learning, but we do not yet understand which assumptions on the distribution or classifier are required. Some experiments also deviate slightly from the predicted behavior. Nevertheless, we believe our conjectures capture the essential aspects of the observed behaviors, at least to first order. It is an important open question to refine these conjectures and better understand their applications and limitations— both theoretically and experimentally.

## 6.1 OPEN QUESTIONS

Our work raises a number of open questions and connections to other areas. We briefly collect some of them here.

1. As described in the limitations, we do not precisely understand the set of distributions and interpolating classifiers for which our conjectures hold. We empirically tested a number of "realistic" settings, but it is open to state formal assumptions defining these settings.

2. It is open to theoretically prove versions of Distributional Generalization for models beyond 1-Nearest-Neighbors. This is most interesting in cases where Distributional Generalization is at odds with classical generalization (e.g. Figure 2A).

3. It is open to understand the mechanisms behind the Agreement Property (Section 4), theoretically or empirically.

4. There are a number of works suggesting "local" behavior of neural networks, and these are somewhat consistent with our locality intuitions in this work. However, it is open to formally understand whether these intuitions are justified in our setting.

5. We give two families of tests $\mathcal{T}$ for which our Interpolating Indistinguishability Meta-conjecture empirically holds. This may not be exhaustive – there may be other ways in which the source distribution $\mathcal{D}$ and test distribution $\mathcal{D}_{\text{te}}$ are close. It is open to explore more ways in which Distributional Generalization holds, beyond the tests presented here.

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

## A    FULL RELATED WORK

Our work is inspired by the broader study of interpolating and overparameterized methods in machine learning; a partial list of works in this theme includes Zhang et al. (2016); Belkin et al. (2018a;b; 2019); Liang and Rakhlin (2018); Nakkiran et al. (2020); Mei and Montanari (2019); Schapire et al. (1998); Breiman (1995); Ghorbani et al. (2019); Hastie et al. (2019); Bartlett et al. (2020); Advani and Saxe (2017); Geiger et al. (2019); Gerace et al. (2020); Chizat and Bach (2020); Goldt et al. (2019); Arora et al. (2019); Allen-Zhu et al. (2019); Neyshabur et al. (2018); Dziugaite and Roy (2017); Muthukumar et al. (2020); Neal et al. (2018).

**Interpolating Methods.** Many of the best-performing techniques on high-dimensional tasks are interpolating methods, which fit their train samples to 0 train error. This includes neural-networks and kernels on images (He et al., 2016; Shankar et al., 2020), and random forests on tabular data (Fernández-Delgado et al., 2014). Interpolating methods have been extensively studied both recently and in the past, since we do not theoretically understand their practical success (Schapire et al., 1998; Schapire, 1999; Breiman, 1995; Zhang et al., 2016; Belkin et al., 2018a;b; 2019; Liang and Rakhlin, 2018; Mei and Montanari, 2019; Hastie et al., 2019; Nakkiran et al., 2020). In particular, much of the classical work in statistical learning theory (uniform convergence, VC-dimension, Rademacher complexity, regularization, stability) fails to explain the success of interpolating methods (Zhang et al., 2016; Belkin et al., 2018a;b; Nagarajan and Kolter, 2019). The few techniques which do apply to interpolating methods (e.g. margin theory (Schapire et al., 1998)) remain vacuous on modern neural-networks and kernels.

**Decision Trees.** In a similar vein to our work, Wyner et al. (2017); Olson and Wyner (2018) investigate decision trees, and show that random forests are equivalent to a Nadaraya–Watson smoother Nadaraya (1964); Watson (1964) with a certain smoothing kernel. Decision trees (Breiman et al., 1984) are often intuitively thought of as "adaptive nearest-neighbors," since they are explicitly a spatial-partitioning method (Hastie et al., 2009). Thus, it may not be surprising that decision trees behave similarly to 1-Nearest-Neighbors. Wyner et al. (2017); Olson and Wyner (2018) took steps towards characterizing and understanding this behavior – in particular, Olson and Wyner (2018) defines an equivalent smoothing kernel corresponding to a random forest, and empirically investigates the quality of the conditional density estimate. Our work presents a formal characterization of the quality of this conditional density estimate (Conjecture 1), which is a novel characterization even for decision trees, as far as we know.

**Kernel Smoothing.** The term kernel regression is sometimes used in the literature to refer to kernel *smoothers*, such as the Nadaraya–Watson kernel smoother (Nadaraya, 1964; Watson, 1964). But in this work we use the term "kernel regression" to refer only to regression in a Reproducing Kernel Hilbert Space, as described in the experimental details.

**Label Noise.** Our conjectures also describe the behavior of neural networks under label noise, which has been empirically and theoretically studied in the past, though not formally characterized before (Zhang et al., 2016; Belkin et al., 2018b; Rolnick et al., 2017; Natarajan et al., 2013; Thulasidasan et al., 2019; Ziyin et al., 2020; Chatterji and Long, 2020). Prior works have noticed that vanilla interpolating networks are sensitive to label noise (e.g. Figure 1 in Zhang et al. (2016), and Belkin et al. (2018b)), and there are many works on making networks more robust to label noise via modifications to the training procedure or objective (Rolnick et al., 2017; Natarajan et al., 2013; Thulasidasan et al., 2019; Ziyin et al., 2020). In contrast, we claim this sensitivity to label noise is not necessarily a problem to be fixed, but rather a consequence of a stronger property: distributional generalization.

**Conditional Density Estimation.** Our density calibration property is similar to the guarantees of a conditional density estimator. More specifically, Conjecture 1 states that an interpolating classifier *samples* from a distribution approximating the conditional density of $p(y|x)$ in a certain sense. Conditional density estimation has been well-studied in classical nonparametric statistics (e.g. the Nadaraya–Watson kernel smoother (Nadaraya, 1964; Watson, 1964)). However, these classical methods behave poorly in high-dimensions, both in theory and in practice. There are some attempts to extend these classical methods to modern high-dimensional problems via augmenting estimators with neural networks (e.g. Rothfuss et al. (2019)). Random forests have also been known to exhibit properties similar to conditional density estimators. This has been formalized in various ways, often only with asymptotic guarantees (Meinshausen, 2006; Pospisil and Lee, 2018; Athey et al., 2019).

No prior work that we are aware of attempts to characterize the quality of the resulting density estimate via testable assumptions, as we do with our formulation of Conjecture 1. Finally, our motivation is not to design good conditional density estimators, but rather to study properties of interpolating classifiers — which we find happen to share properties of density estimators.

**Uncertainty and Calibration.** The Agreement Property (Conjecture 2) bears some resemblance to uncertainty estimation (e.g. Lakshminarayanan et al. (2017)), since it estimates the the test error of a classifier using an ensemble of 2 models trained on disjoint train sets. However, there are important caveats: (1) Our Agreement Property only holds on-distribution, and degrades on off-distribution inputs. Thus, it is not as helpful to estimate out-of-distribution errors. (2) It only gives an estimate of the average test error, and does not imply pointwise calibration estimates for each sample.

Feature Calibration (Conjecture 1) is also related to the concepts of calibration and multicalibration (Guo et al., 2017; Niculescu-Mizil and Caruana, 2005; Hébert-Johnson et al., 2018). In our framework, calibration is implied by Feature Calibration for a specific set of partitions $L$ (determined by level sets of the classifier's confidence). However, we are not concerned with a specific set of partitions (or "subgroups" in the algorithmic fairness literature) but we generally aim to characterize for which partitions Feature Calibration holds. Moreover, we consider only hard-classification decisions and not confidences, and we study only standard learning algorithms which are not given any distinguished set of subgroups/partitions in advance. Our notion of distributional generalization is also related to the notion of "distributional subgroup overfitting" introduced recently by Yaghini et al. (2019) to study algorithmic fairness. This can be seen as studying distributional generalization for a specific family of tests (determined by distinguished subgroups in the population).

**Locality and Manifold Learning.** Our intuition for the behaviors in this work is that they arise due to some form of "locality" of the trained classifiers, in an appropriate space. This intuition is present in various forms in the literature, for example: the so-called called "manifold hypothesis," that natural data lie on a low-dimensional manifold (e.g. Narayanan and Mitter (2010); Sharma and Kaplan (2020)), as well as works on local stiffness of the loss landscape (Fort et al., 2019), and works showing that overparameterized neural networks can learn hidden low-dimensional structure in high-dimensional settings (Gerace et al., 2020; Bach, 2017; Chizat and Bach, 2020). It is open to more formally understand connections between our work and the above.

## B   EXPERIMENTAL DETAILS

Here we describe general background, and experimental details common to all sections. Then we provide section-specific details below.

### B.1   DATASETS

We consider the image datasets CIFAR-10 and CIFAR-100 (Krizhevsky et al., 2009), MNIST (LeCun et al., 1998), Fashion-MNIST (Xiao et al., 2017), CelebA (Liu et al., 2015), and ImageNet (Russakovsky et al., 2015). We normalize images to $x \in [0, 1]^{C \times W \times H}$.

We also consider tabular datasets from the UCI repository Dua and Graff (2017). For UCI data, we consider the 121 classification tasks as standardized in Fernández-Delgado et al. (2014). Some of these tasks have very few examples, so we restrict to the 92 classification tasks from Fernández-Delgado et al. (2014) which have at least 200 total examples.

### B.2   MODELS

We consider neural-networks, kernel methods, and decision trees.

#### B.2.1   DECISION TREES

We train interpolating decision trees using a growth rule from Random Forests (Breiman, 2001; Ho, 1995): selecting a split based on a random $\sqrt{d}$ subset of $d$ features, splitting based on Gini impurity, and growing trees until all leafs have a single sample. This is as implemented by Scikit-learn Pedregosa et al. (2011) defaults with `RandomForestClassifier(n_estimators=1, bootstrap=False)`.

### B.2.2 KERNELS

Throughout this work we consider classification via kernel regression and kernel SVM. For $M$-class classification via kernel regression, we follow the methodology in e.g. Rahimi and Recht (2008); Belkin et al. (2018b); Shankar et al. (2020). We solve the following convex problem for training:

$$\alpha^* := \operatorname*{argmin}_{\alpha \in \mathbb{R}^{N \times M}} ||K\alpha - y||_2^2 + \lambda \alpha^T K \alpha$$

where $K_{ij} = k(x_i, x_j)$ is the kernel matrix of the training points for a kernel function $k$, $y \in \mathbb{R}^{N \times M}$ is the one-hot encoding of the train labels, and $\lambda \geq 0$ is the regularization parameter. The solution can be written

$$\alpha^* = (K + \lambda I)^{-1} y$$

which we solve numerically using SciPy `linalg.solve` (Virtanen et al., 2020). We use the explicit form of all kernels involved. That is, we do not use random-feature approximations (Rahimi and Recht, 2008), though we expect they would behave similarly.

The kernel predictions on test points are then given by

$$g_\alpha(x) := \sum_{i \in [N]} \alpha_i k(x_i, x) \tag{8}$$

$$f_\alpha(x) := \operatorname*{argmax}_{j \in [M]} g_\alpha(x)_j \tag{9}$$

where $g(x) \in \mathbb{R}^M$ are the kernel regressor outputs, and $g(x) \in [M]$ is the thresholded classification decision. This is equivalent to training $M$ separate binary regressors (one for each label), and taking the argmax for classification. We usually consider *unregularized* regression ($\lambda = 0$), except in Section 5.

For kernel SVM, we use the implementation provided by Scikit-learn (Pedregosa et al., 2011) `sklearn.svm.SVC` with a precomputed kernel, for inverse-regularization parameter $C \geq 0$ (larger $C$ corresponds to smaller regularization).

**Types of Kernels.** We use the following kernel functions $k : \mathbb{R}^d \times \mathbb{R}^d \to \mathbb{R}_{\geq 0}$.

- Gaussian Kernel (RBF): $k(x_i, x_j) = \exp(-\frac{||x_i - x_j||_2^2}{2\widetilde{\sigma}^2})$.
- Laplace Kernel: $k(x_i, x_j) = \exp(-\frac{||x_i - x_j||_2}{\widetilde{\sigma}})$.
- Myrtle10 Kernel: This is the compositional kernel introduced by Shankar et al. (2020). We use their exact kernel for CIFAR-10.

For the Gaussian and Laplace kernels, we parameterize bandwidth by $\sigma := \widetilde{\sigma}/\sqrt{d}$. We use the following bandwidths, found by cross-validation to maximize the unregularized test accuracy:

- MNIST: $\sigma = 0.15$ for RBF kernel.
- Fashion-MNIST: $\sigma = 0.1$ for RBF kernel. $\sigma = 1.0$ for Laplace kernel.
- CIFAR-10: Myrtle10 Kernel from Shankar et al. (2020), and $\sigma = 0.1$ for RBF kernel.

### B.2.3 NEURAL NETWORKS

We use 4 different neural networks in our experiments. We use a multi-layer perceptron, and three different Residual networks.

**MLP:** We use a Multi-layer perceptron or a fully connected network with 3 hidden layers with 512 neurons in each layer. A hidden layer is followed by a BatchNormalization layer and ReLU activation function.

**WideResNet:** We use the standard WideResNet-28-10 described in Zagoruyko and Komodakis (2016). Our code is based on this repository.

**ResNet50:** We use a standard ResNet-50 from the PyTorch library (Paszke et al., 2017).

|  | MLP | ResNet18 | WideResNet | ResNet50 |
|---|---|---|---|---|
| **Batchsize** | 128 | 128 | 128 | 32 |
| **Epochs** | 820 | 200 | 200 | 50 |
| **Optimizer** | Adam ($\beta_1 = 0.9, \beta_2 = 0.999$) | SGD + Momentum (0.9) | SGD + Momentum (0.9) | SGD |
| **Learning rate** (LR) schedule | Constant LR = 0.001 | Inital LR= 0.05 scale by 0.1 at epochs $(80, 120)$ | Inital LR= 0.1 scale by 0.2 at epochs $(80, 120, 160)$ | Initial LR = 0.001, scale by 0.1 if training loss stagnant for 2000 gradient steps |
| **Data Augmentation** | Random flips + RandomCrop(32, padding=4) | | | |
| **CIFAR-10 Error** | $\sim 37\%$ | $\sim 8\%$ | $\sim 4\%$ | N/A |

Table 1: Hyperparameters used to train the neural networks and their errors on the unmodified CIFAR-10 dataset

**ResNet18:** We use a modification of ResNet18 He et al. (2016) adapted to CIFAR-10 image sizes. Our code is based on this repository.

For Experiment 1 and Section 3, the hyperparameters used to train the above networks are given in Table 1.

## C   FEATURE CALIBRATION: APPENDIX

### C.1   A GUIDE TO READING THE PLOTS

All the experiments in support of Conjecture 1 (experiments in Section 3 and the Introduction) involve various quantities which we enumaerate here

1. Inputs $x$: Each experiment involves inputs from a standard dataset like CIFAR-10 or MNIST. We use the standard train/test splits for every dataset.

2. Distinguishable feature $L(x)$: This feature depends only on input $x$. We consider various features like the original classes itself, a superset of classes (as in coarse partition) or some secondary attributes (like the binary attributes provided with CelebA)

3. Output labels $y$: The output label may be some modification of the original labels. For instance, by adding some type of label noise, or a constructed binary task as in Experiment 1

4. Classifier family $F$: We consider various types of classifiers like neural networks trained with gradient based methods, kernel and decision trees.

In each experiment, we are interested in two joint densities $(y, L(x))$, which depends on our dataset and task and is common across train and test, and $(f(x), L(x))$ which depends on the interpolating classifiers outputs on the *test* set. Since $y, L(x)$ and $f(x)$ are discrete, we will look at their discrete joint distributions. We sometimes refer to $(y, L(x))$ as the train joint density, as at interpolation $(y, L(x)) = (f(x), L(x))$ for all training inputs $x$. We also refer to $(f(x), L(x))$ as the test density, as we measure this only on the test set.

### C.2   EXPERIMENT 1

**Experimental details:** We now provide further details for Experiment 1. We first construct a dataset from CIFAR-10 that obeys the joint density $(y, L(x))$ shown in Figure 1 left panel. We then train a WideResNet-28-10 (WRN-28-10) on this modified dataset to zero training error. The network is trained with the hyperparameters described in Table 1. We then observe the joint density $(f(x), L(x))$ on the test images and find that the two joint densities are close as shown in Figure 5.

We now consider a modification of this experiment as follows:

**Experiment 2.** *Consider the following distribution over images $x$ and binary labels $y$. Sample $x$ as a uniformly random CIFAR-10 image, and sample the label as $p(y|x) = Bernoulli(\texttt{CIFAR\_Class(x)}/10)$. That is, if the CIFAR-10 class of $x$ is $k \in \{0, 1, \ldots 9\}$, then the label is $1$ with probability $(k/10)$ and $0$ otherwise. Figure 5 shows this joint distribution of $(x, y)$. As before, train a WideResNet to $0$ training error on this distribution.*

In this experiment too, we observe that the train and test joint densities are close as shown in Figure 5.

Now, we repeat the same experiment, but with an MLP instead of WRN-28-10. The training procedure is described in Table 1. This MLP has an error on $37\%$ on the original CIFAR-10 dataset.

Since this MLP has poor accuracy on the original CIFAR-10 classification task, it does not form a distinguishable partition for it. As a result, the train and test joint densities (Figure 6) do not match as well as they did for WRN-28-10.

### C.3   CONSTANT PARTITION

Consider the trivially-distinguishable *constant* feature $L(x) = 0$. Then, Conjecture 1 states that the marginal distribution of class labels for any interpolating classifier $f(x)$ is close to the true marginals $p(y)$. That is, irrespective of the classifier's test accuracy, it outputs the "right" proportion of class labels on the test set, even when there is strong class imbalance.

To show this, we construct a dataset based on CIFAR-10 that has class-imbalance. For class $k \in \{0...9\}$, sample $(k + 1) \times 500$ images from that class. This will give us a dataset where classes will have marginal distribution $p(y = \ell) \propto \ell + 1$ for classes $\ell \in [10]$, as shown in Figure 7. We do this both for the training set and the test set, to keep the distribution $\mathcal{D}$ fixed.

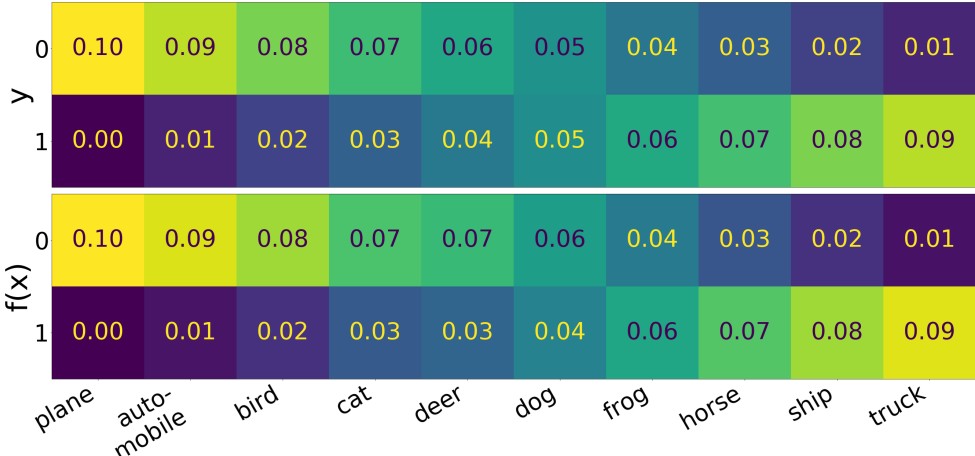

Figure 5: **Distributional Generalization in Experiment 2.** Joint densities of the distributions involved in Experiment 2. The top panel shows the joint density of labels on the train set: $(\texttt{CIFAR\_Class(x)}, y)$. The bottom panels shows the joint density of classifier predictions on the test set: $(\texttt{CIFAR\_Class(x)}, f(x))$. Distributional Generalization claims that these two joint densities are close.

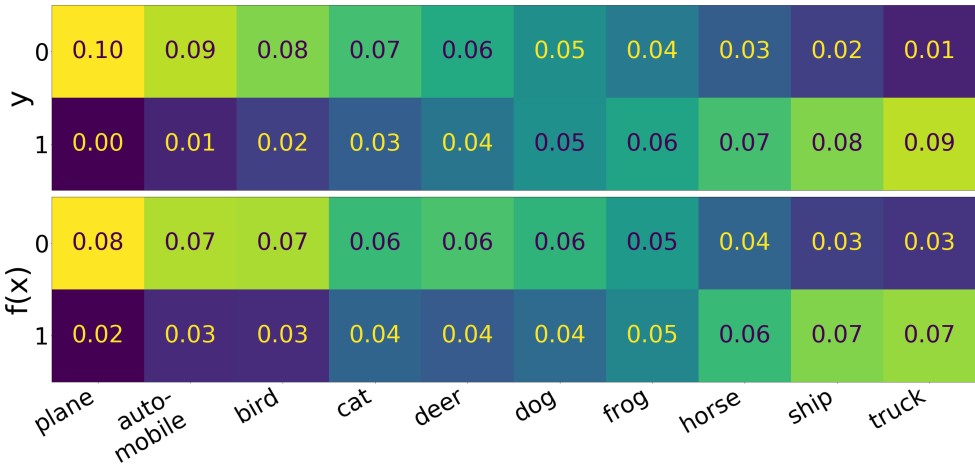

Figure 6: Joint density of $(y, \text{Class}(x))$, top, and $(f(x), \text{Class}(x))$, bottom, for test samples $(x, y)$ from Experiment 2 for an MLP.

We then train a variety of classifiers (MLPs, Kernels, ResNets) to interpolation on this dataset, which have varying levels of test errors (9-41%). The class balance of classifier outputs on the (rebalanced) test set is then close to the class balance on the train set, irrespective of the classifier error. Full experimental details and results are described in Appendix C. Note that a 1-nearest-neighbors classifier would have this property.

## C.4 CLASS PARTITION

### C.4.1 NEURAL NETWORKS AND CIFAR-10

We now provide experiments in support of Conjecture 1 when the class itself is a distinguishable partition. A WRN-28-10 achieves an error of ~4% on CIFAR-10. Hence, the original labels in CIFAR-10 form a distinguishable partition for this dataset. To demonstrate that Conjecture 1 holds, we consider different structured label noise on the CIFAR-10 dataset. To do so, we apply a variety of confusion matrices to the data. That is, for a confusion matrix $C : 10 \times 10$ matrix, the element $c_{ij}$

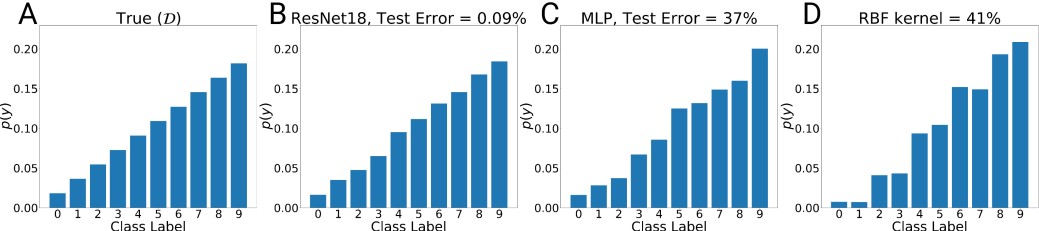

Figure 7: **Feature Calibration for Constant Partition** $L$**:** The CIFAR-10 train and test sets are class-rebalanced according to (A). Interpolating classifiers are trained on the train set, and we plot the class-balance of their outputs on the test set. This roughly matches the class-balance of the train set, even for poorly-generalizing classifiers.

gives the joint density that a randomly sampled image had original label $j$, but is flipped to class $i$. For no noise, this would be an identity matrix.

We begin by a simple confusion matrix where we flip only one class $0 \to 1$ with varying probability $p$. Figure **??** shows one such confusion matrix for $p = 0.4$. We then train a WideResNet-28-10 to zero train error on this dataset. We use the hyperparameters described in B.2 We find that the classifier outputs on the test set closely track the confusion matrix that was applied to the distribution. Figure 8 shows that this is independent of the value of $p$ and continues to hold for $p = [0, 1]$.

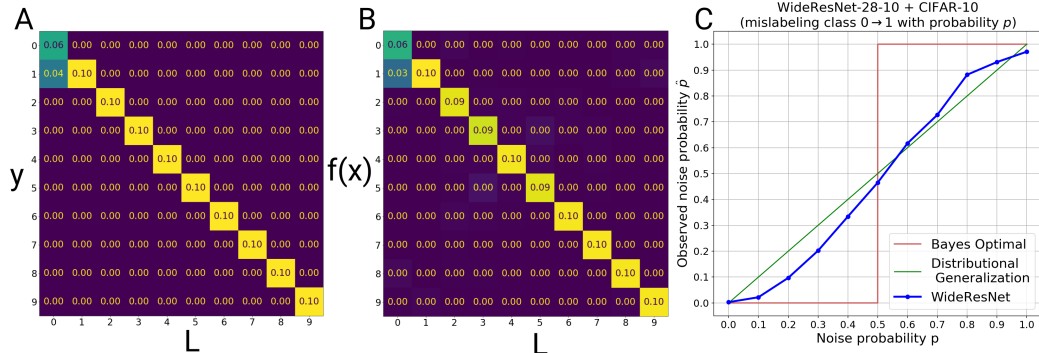

Figure 8: **Feature Calibration with original classes on CIFAR-10**: We train a WRN-28-10 on the CIFAR-10 dataset where we mislabel class $0 \to 1$ with probability $p$. (A): Joint density of the distinguishable features $L$ (the original CIFAR-10 class) and the classification task labels $y$ on the train set for noise probability $p = 0.4$. (B): Joint density of the original CIFAR-10 classes $L$ and the network outputs $f(x)$ on the test set. (C): Observed noise probability in the network outputs on the test set (the $(1, 0)$ entry of the matrix in B) for varying noise probabilities $p$

To show that this is not dependent on the particular class used, we also show that the same holds for a random confusion matrix. We generate a sparse confusion matrix as follows. We set the diagonal to $0.5$. Then, for every class $j$, we pick any two random classes for and set them to $0.2$ and $0.3$. We train a WRN-28-10 on it and report the test confusion matrix. The resulting train and test densities are shown in Figure 9. As expected, the train and test confusion matrices are close, and share the same sparsity pattern.

#### C.4.2 DECISION TREES

Figure 10 shows a version of this experiment for decision trees on the molecular biology UCI task. The molecular biology task is a 3-way classification problem: to classify the type of a DNA splice junction (donor, acceptor, or neither), given the sequence of DNA (60 bases) surrounding the junction. We add varying amounts of label noise that flips class 2 to class 1 with a certain probability, and we observe that interpolating decision trees reproduce this same structured label noise on the test set.

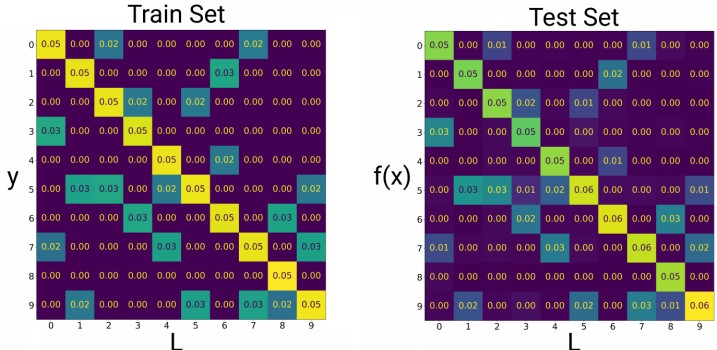

Figure 9: **Feature Calibration with random confusion matrix on CIFAR-10:** Left: Joint density of labels $y$ and original class $L$ on the train set. Right: Joint density of classifier predictions $f(x)$ and original class $L$ on the test set, for a WideResNet28-10 trained to interpolation. These two joint densities are close, as predicted by Conjecture 1.

Similar results hold for decision trees; here we show experiments on two UCI tasks: `wine` and `mushroom`.

The `wine` task is a 3-way classification problem: to identify the cultivar of a given wine (out of 3 cultivars), given 13 physical attributes describing the wine. Figure 11 shows an analogous experiment with label noise taking class 1 to class 2.

The `mushroom` task is a 2-way classification problem: to classify the type of edibility of a mushroom (edible vs poisonous) given 22 physical attributes (e.g. stalk color, odor, etc). Figure 12 shows an analogous experiment with label noise flipping class 0 to class 1.

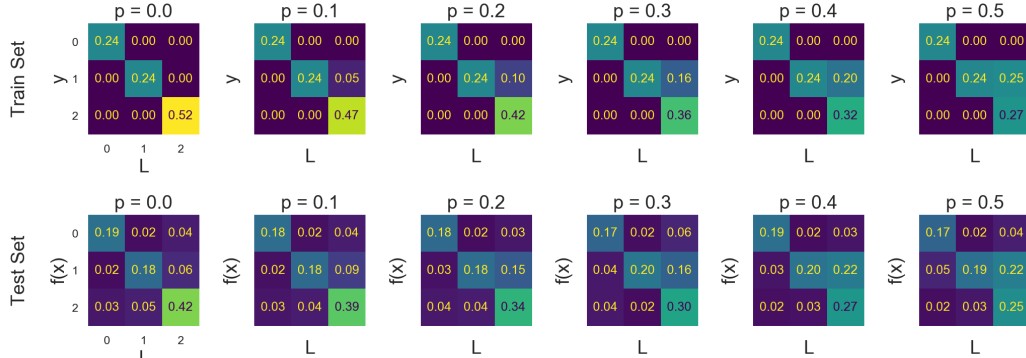

Figure 10: **Feature Calibration for Decision trees on UCI (molecular biology).** We add label noise that takes class 2 to class 1 with probability $p \in [0, 0.5]$. The top row shows the confusion matrix of the true class $L(x)$ vs. the label $y$ on the train set, for varying levels of noise $p$. The bottom row shows the corresponding confusion matrices of the classifier predictions $f(x)$ on the test set, which closely matches the train set, as predicted by Conjecture 1.

## C.5 MULTIPLE FEATURES

Conjecture 1 states that the network should be automatically calibrated for all distinguishable features, without any explicit labels for them. To verify this, we use the CelebA dataset (Liu et al., 2015), containing images with various labelled binary attributes per-image ("male", "blond hair", etc). Some of these attributes form a distinguishable feature for ResNet50 as they are learnable to high accuracy (Jahandideh et al., 2018). We pick one of hard attributes as the target classification task. We train a ResNet-50 to predict the attribute {Attractive, Not Attractive}. We choose this attribute because a ResNet-50 performs poorly on this task (test error $\sim 20\%$) and has good class balance. We choose

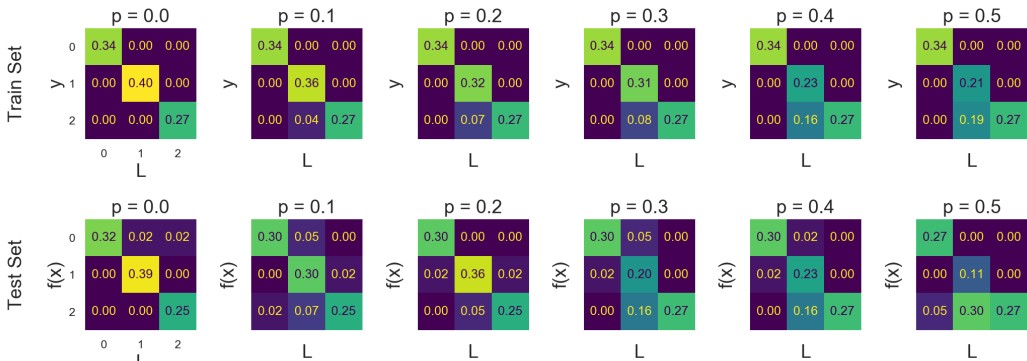

Figure 11: Decision trees on UCI (wine). We add label noise that takes class 1 to class 2 with probability $p \in [0, 0.5]$. Each column shows the test and train confusion matrices for a given $p$. Note that this decision trees achieve high accuracy on this task with no label noise (leftmost column). We plot the empirical joint density of the train set, and not the population joint density of the train distribution, and thus the top row exhibits some statistical error due to small-sample effects.

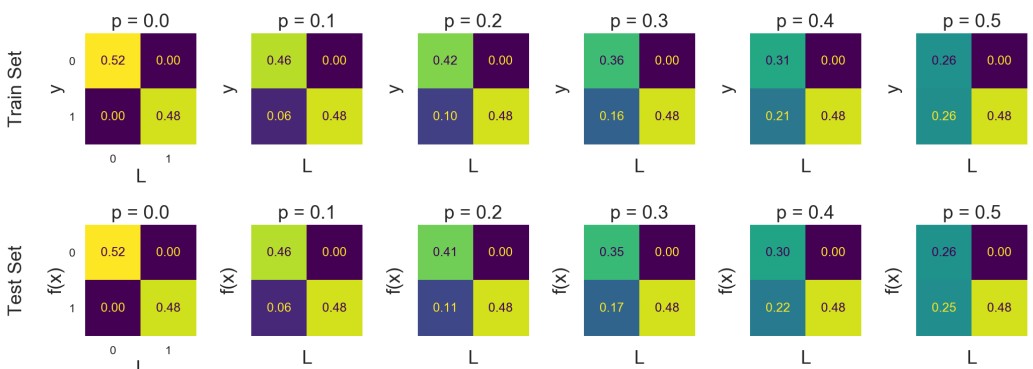

Figure 12: Decision trees on UCI (mushroom). We add label noise that takes class 0 to class 1 with probability $p \in [0, 0.5]$. Each column shows the test and train confusion matrices for a given $p$. Note that this decision trees achieve high accuracy on this task with no label noise (leftmost column).

an attribute with poor generalization because the conjecture would hold trivially for if the network generalizes well. We initialize the network with a pretrained ResNet-50 from the PyTorch library Paszke et al. (2017) and use the hyperparameters described in Section B.2 to train on this attribute. We then check the train/test joint density with various other attributes like Male, Wearing Lipstick etc. Note that the network is not given any label information for these additional attributes, but is calibrated with respect to them. That is, the network says $\sim 30\%$ of images that have 'heavy makeup' will be classified as 'Attractive', even if the network makes mistakes on which particular inputs it chooses to do so. In this setting, the label distribution is deterministic, and not directly dependent on the distinguishable features, unlike the experiments considered before. Yet, as we see in Figure 13, the classifier outputs are correctly calibrated for each attribute. Loosely, this can be viewed as the network performing 1NN classification in a metric space that is well separated for each of these distinguishable features.

## C.6 COARSE PARTITION

We now consider cases where the original classes do not form a distinguishable partition for the classifier in consideration. That is, the classifier is not powerful enough to obtain low error on the original dataset, but can perform well on a coarser division of the classes.

To verify this, we consider a division of the CIFAR-10 classes into Objects {airplane, automobile, ship, truck} vs Animals {cat, deer, dog, frog}. An MLP trained on this problem has low error ($\sim 8\%$),

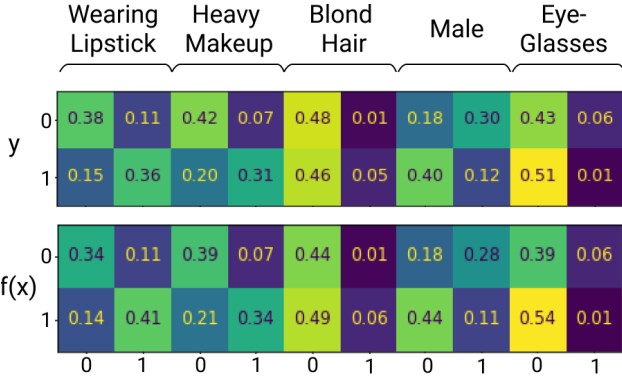

Figure 13: **Feature Calibration for multiple features on CelebA**: We train a ResNet-50 to perform binary classification task on the CelebA dataset. The top row shows the joint distribution of this task label with various other attributes in the dataset. The bottom row shows the same joint distribution for the ResNet-50 outputs on the test set. Note that the network was not given any explicit inputs about these attributes during training.

| Model | AlexNet | ResNet18 | ResNet50 | BagNet8 | BagNet32 |
|---|---|---|---|---|---|
| ImageNet Accuracy | 0.565 | 0.698 | 0.761 | 0.464 | 0.667 |
| Accuracy on dogs | 0.588 | 0.729 | 0.793 | 0.462 | 0.701 |
| Accuracy on terriers | 0.572 | 0.704 | 0.775 | 0.421 | 0.659 |
| Accuracy for binary {dog/not-dog} | 0.984 | 0.993 | 0.996 | 0.972 | 0.992 |
| Accuracy on {terrier/not-terrier} among dogs | 0.913 | 0.955 | 0.969 | 0.876 | 0.944 |
| Fraction of real-terriers among dogs | 0.224 | 0.224 | 0.224 | 0.224 | 0.224 |
| **Fraction of predicted-terriers among dogs** | 0.209 | 0.222 | 0.229 | 0.192 | 0.215 |

Table 2: ImageNet classifiers are calibrated with respect to dogs: All classifiers predict terrier for roughly $\sim 22\%$ of all dogs (last row), though they may mistake which specific dogs are terriers.

but the same network performs poorly on the full dataset ($\sim 37\%$ error). Hence, Object vs Animals forms a distinguishable partition with MLPs. In Figure 14a, we show the results of training an MLP on the original CIFAR-10 classes. We see that the network mostly classifies objects as objects and animals as animals, even when it might mislabel a dog for a cat.

We perform a similar experiment for the RBF kernel on Fashion-MNIST, with partition {clothing, shoe, bag}, in Figure 14b.

**ImageNet experiment.** In Table 2 we provide results of the terrier experiment in the body, for various ImageNet classifiers. We use publicly available pretrained ImageNet models from this repository, and use their evaluations on the ImageNet test set.

## C.7 QUANTITATIVE PREDICTIONS: TV DISTANCE VS $\varepsilon$

We now test the quantitative predictions made by Conjecture 1. This conjecture states that the TV-distance between the joint distributions $(L(x), f(x))$ and $(L(x), y)$ is at most $\varepsilon$, where $\varepsilon$ is the error of the training procedure in learning $L$ (see Definition 1).

$$TV((L(x), f(x)), (L(x), y)) \le \varepsilon$$

To test this, we consider binary task similar to Experiment 1 where (Ship, Plane) are labeled as class 0 and (Cat, Dog) are labeled as class 1. We then add add noise to $p = 0.3$ fraction of cats to mislabel them as class 0. Then, we train a convolutional network to interpolation on this task. In this experiment, $(Cat, Dog, Plane, Ship)$ form distinguishable features. To vary the error $\varepsilon$ on these

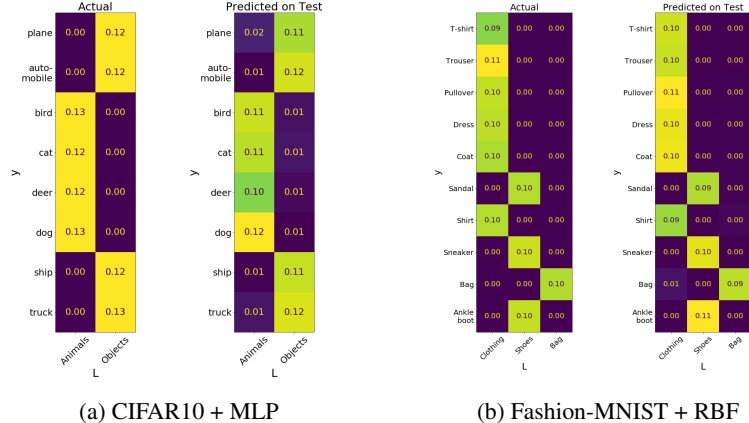

(a) CIFAR10 + MLP          (b) Fashion-MNIST + RBF

Figure 14: Coarse partitions as distinguishable features: We consider a setting where the original classes are not distinguishable, but the a superset of the classes are.

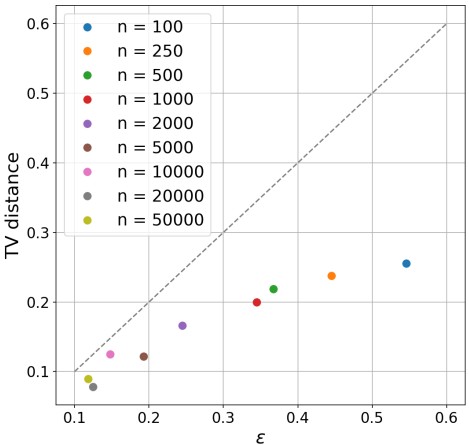

Figure 15: TV distance between $(L(x), f(x))$ and $(L(x), y)$ vs. $\varepsilon$ for the distinguishable features

distinguishable features systematically, we train a networks to classify $(Cat, Dog, Plane, Ship)$ with changing number of samples. Networks with fewer samples have larger $\varepsilon$ since they are worse at classifying the distinguishable features. Then, we use the same setup and number of samples respectively to train networks on the binary task and measure the TV-distance between $(L(x), f(x))$ and $(L(x), y)$ in this task. The results are shown in Figure 15. As predicted, the TV distance is upper bounded by $\varepsilon$.

## C.8    DISCUSSION: PROPER SCORING RULES

Here we distinguish the density-estimation of Conjecture 1 from another setting where density estimation occurs. If $\ell(\widehat{p}, y)$ is a *strictly-proper scoring rule*[3] on predicted distribution $\widehat{p} \in \Delta(\mathcal{Y})$ and sample $y \in \mathcal{Y}$, then the population minimizer of $\ell(F(x), y)$ is exactly the conditional density $F(x) = p(y|x)$. That is,

$$p(y|x) = \underset{F:\mathcal{X}\to\Delta(\mathcal{Y})}{\operatorname{argmin}} \ \underset{(x,y)\sim p}{\mathbb{E}} \left[ \ell(F(x), y) \right]$$

---

[3] See Gneiting and Raftery (2007) for a survey of proper scoring rules.

This suggests that in the limit of large-capacity network and very large data (to approximate population quantities), training neural nets with cross-entropy loss on samples $(x, y)$ will yield a good density estimate of $p(y|x)$ at the softmax layer. However, this is not what is happening in our experiments. First, our experiments consider the hard-thresholded classifier, i.e. the argmax of the softmax layer. If the softmax layer itself was close to $p(y|x)$, then the classifier itself will be close to $\text{argmax}_y\, p(y|x)$ – that is, close to the optimal classifier. However, this is not the case (since the classifiers make significant errors). Second, we observe Conjecture 1 even in settings where we train with non-proper scoring rules (e.g. kernel regression, where the classifier does not output a probability).

## D    AGREEMENT PROPERTY: APPENDIX

### D.1    EXPERIMENTAL DETAILS

For ResNets on CIFAR-10 and CIFAR-100, we use the following training procedure. For $n \leq 25000$, we sample two disjoint train sets $S_1, S_2$ of size $n$ from the 50K total train samples. Then we train two ResNet18s $f_1, f_2$ on $S_1, S_2$ respectively. We optimize using SGD on the cross-entropy loss, with batch size 128, using learning rate schedule 0.1 for $40\lfloor \frac{50000}{n} \rfloor$ epochs, then 0.01 for $20\lfloor \frac{50000}{n} \rfloor$ epochs. That is, we scale up the number of epoches for smaller train sizes, to keep the number of gradient steps constant. We also early-stop optimization when the train loss reaches $< 0.0001$, to save computational time. For experiments with data-augmentation, we use horizontal flips and `RandomCrop(32, padding=4)`. We estimate test accuracy and agreement probability on the CIFAR-10/100 test sets.

For the kernel experiments on Fashion-MNIST, we repeat the same procedure: we sample two disjoint train sets from all the train samples, train kernel regressors, and evaluate their agreement on the test set. Each point on the figures correspond to one trial.

For UCI, some UCI tasks have very few examples, and so here we consider only the 92 classification tasks from Fernández-Delgado et al. (2014) which have at least 200 total examples. For each task, we randomly partition all the examples into a 40%-40%-20% split for 2 disjoint train sets, and 1 test set (20%). We then train two interpolating decision trees, and compare their performance on the test set. Decision trees are trained using a growth rule from Random Forests (Breiman, 2001; Ho, 1995): selecting a split based on a random $\sqrt{d}$ subset of $d$ features, splitting based on Gini impurity, and growing trees until all leafs have a single sample. This is as implemented by Scikit-learn Pedregosa et al. (2011) defaults with `RandomForestClassifier(n_estimators=1, bootstrap=False)`.

### D.2    ADDITIONAL PLOTS

Figure 16 shows the Agreement Property for the Myrtle Kernel on CIFAR-10, and Figure 17 shows the RBF and Laplace kernels on Fashion-MNIST.

Figure 18a shows interpolating decision trees on 90 classification UCI tasks. We plot the means of agreement probability and test accuracy when averaged over 100 random partitions for each task, and also include the corresponding plot for a single trial. Error bars on UCI tasks show 95% Clopper-Pearson confidence intervals in estimating population quantities.

### D.3    POTENTIAL MECHANISMS

We now consider, and refute, several potential mechanisms which could explain the experimental results of Conjecture 2 ("Agreement Property"). In this section, we would like to understand the distribution of $f \leftarrow \text{Train}(\mathcal{D}^n)$, in order to evaluate several proposed mechanisms. Technically, sampling from this distribution requires training a classifier on a *fresh* train set. Since we do not have infinite samples for CIFAR-10, we construct empirical estimates by training an ensemble of classifiers on random subsets of CIFAR-10. Then, to approximate a sample $f \leftarrow \text{Train}(\mathcal{D}^n)$, we simply sample from our ensemble $f \leftarrow \{f_i\}_i$.

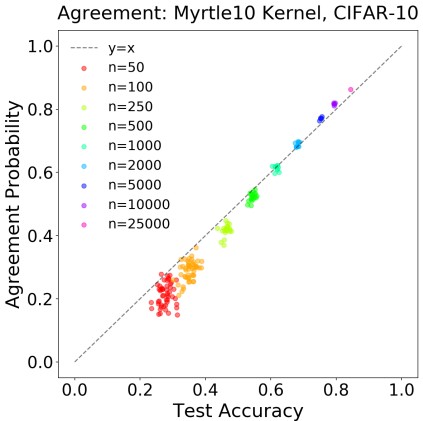

Figure 16: **Agreement Property for Myrtle Kernel on CIFAR-10.**

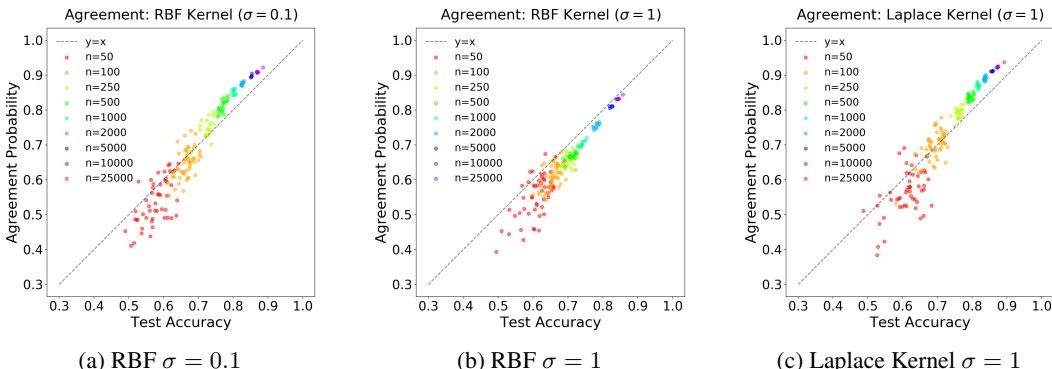

(a) RBF $\sigma = 0.1$        (b) RBF $\sigma = 1$        (c) Laplace Kernel $\sigma = 1$

Figure 17: **Agreement Property for Kernels on Fashion-MNIST.** For two classifiers trained on disjoint train sets, the probability they agree with each other (on the test set) is close to their test accuracy.

### D.3.1 BIMODAL SAMPLES

A simple model which would exhibit the Agreement Property is the following: Suppose test samples $x$ come in two types: "easy" or "hard." All classifiers get "easy" samples correct, but they output a uniformly random class on "hard" samples. That is, for a fixed $x$, consider the probability that a freshly-trained classifier gets $x$ correct. "Easy" samples are such that

$$\text{For } x \in \text{EASY:} \quad \Pr_{f \leftarrow \text{Train}(\mathcal{D}^n)}[f(x) = y(x)] = 1$$

while "hard" samples have a uniform distribution on output classes $[K]$:

$$\text{For } x \in \text{HARD:} \quad \Pr_{f \leftarrow \text{Train}(\mathcal{D}^n)}[f(x) = i] = \frac{1}{K} \quad \forall i \in [K]$$

Notice that for HARD samples $x$, a classifier $f_1$ agrees with the true label $y$ with exactly the same probability that it agrees with an independent classifier $f_2$ (because both $f_1, f_2$ are uniformly random on $x$). Thus, the agreement property (Conjecture 2) holds exactly under this model. However, this strict decomposition of samples into "easy" and "hard" does not appear to be the case in the experiments, as shown below.

Figure 19 shows a histogram of

$$h(x) := \Pr_{f \leftarrow \text{Train}(\mathcal{D}^n)}[f(x) = y]$$

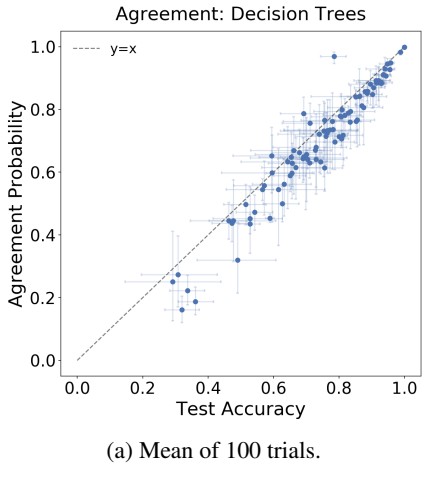

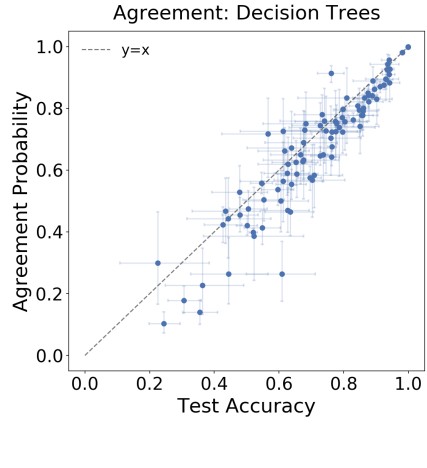

(a) Mean of 100 trials.

(b) Single trial.

Figure 18: **Agreement Property on UCI.** For two decision trees trained on disjoint train sets, the probability they agree with each other (on the test set) is close to their test accuracy. Each point corresponds to one UCI task, and error bars show 95% Clopper-Pearson confidence intervals in estimating population quantities.

for test samples $x$ in CIFAR-10, where $f$ is a ResNet18 trained on 5000 samples[4]. This quantity can be interpreted as the "easiness" of a given test sample $(x, y)$ to a certain classifier family.

If the EASY/HARD bimodal model were true, we would expect the distribution of $h(x)$ to be concentrated on $h(x) = 1$ (easy samples) and $h(x) = 0.1$ (hard samples). But this is not the case in Figure 19, and thus the bimodal model does not explain the Agreement Property.

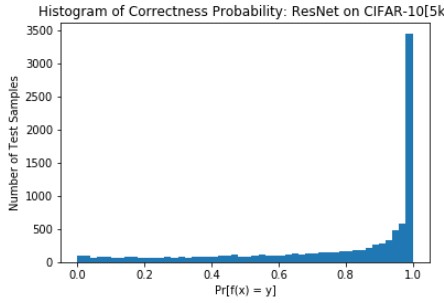

Figure 19: Histogram of sample-hardnesses.      Figure 20: Pointwise agreement histogram.

### D.3.2   POINTWISE AGREEMENT

We could more generally posit that Conjecture 2 is true because the Agreement Property holds *pointwise* for most test samples $x$:

$$\text{w.h.p. for } (x, y) \sim \mathcal{D}: \quad \Pr_{f_1 \leftarrow \text{Train}(\mathcal{D}^n)} [f_1(x) = y] \approx \Pr_{\substack{f_1 \leftarrow \text{Train}(\mathcal{D}^n) \\ f_2 \leftarrow \text{Train}(\mathcal{D}^n)}} [f_1(x) = f_2(x)] \quad (10)$$

However, we find (perhaps surprisingly) that this is not the case. To see why this is surprising, observe that Conjecture 2 implies that the agreement probability is close to test accuracy, *in expectation* over

---

[4]We estimate this probability over the empirical ensemble $f \leftarrow \{f_i\}_i$, where each $f_i$ is a classifier trained on a random $5k$-subset of CIFAR-10. We train 100 classifiers in this ensemble.

the test sample and the classifiers $f_1, f_2 \leftarrow \text{Train}(\mathcal{D}^n)$:

$$\Pr_{\substack{f_1 \\ (x,y)\sim\mathcal{D}}}[f_1(x) = y] \approx \Pr_{\substack{f_1,f_2 \\ (x,y)\sim\mathcal{D}}}[f_1(x) = f_2(x)] \qquad \text{(Conjecture 2)}$$

$$\iff \underset{f_1}{\mathbb{E}} \underset{x,y\sim\mathcal{D}}{\mathbb{E}}[\mathbb{1}\{f_1(x) = y\}] \approx \underset{f_1,f_2}{\mathbb{E}} \underset{x,y\sim\mathcal{D}}{\mathbb{E}}[\mathbb{1}\{f_1(x) = f_2(x)\}] \tag{11}$$

Swapping the order of expectation, this implies

$$\underset{x,y\sim\mathcal{D}}{\mathbb{E}} \underbrace{\left[ \underset{f_1,f_2}{\mathbb{E}}[\mathbb{1}\{f_1(x) = y\} - \mathbb{1}\{f_1(x) = f_2(x)\}] \right]}_{M(x,y)} \approx 0 \tag{12}$$

Now, we may expect that this means $M(x, y) \approx 0$ pointwise, for most test samples $(x, y)$. But this is not the case. It turns out that $M(x, y)$ takes on significantly positive and negative values, and these effects "cancel out" in expectation over the distribution, to yield Conjecture 2.

For example, we compute $M(x, y)$ for the Myrtle10 kernel on CIFAR-10 with 1000 train samples. [5]

1. The agreement probability is within $0.8\%$ of the test error (as in Figure 16), and so

$$\underset{x,y\sim\mathcal{D}}{\mathbb{E}}[M(x,y)] \approx 0.008$$

2. However, $M(x, y)$ is not pointwise close to 0. E.g,

$$\underset{x,y\sim\mathcal{D}}{\mathbb{E}}[|M(x,y)|] \approx 0.133$$

Figure 20 plots the distribution of $M(x, y)$. We see that some samples $(x, y)$ have high agreement probability, and some low, and these happen to balance in expectation to yield the test accuracy.

Interestingly, 1-nearest neighbors can satisfy the agreement property of Claim 2 without satisfying the "pointwise agreement" of Equation 10. It remains an open problem to understand the mechanisms behind the Agreement Property.

## E   NEAREST-NEIGHBOR PROOFS

### E.1   FEATURE CALIBRATION PROPERTY

*Proof of Theorem 1.* Recall that $L$ being an $(\varepsilon, \text{NN}, \mathcal{D}, n)$-distinguishable partition means that nearest-neighbors works to classify $L(x)$ from $x$:

$$\Pr_{\substack{\{x_i,y_i\}\sim\mathcal{D}^n \\ S=\{(x_i,L(x_i)\} \\ x,y\sim\mathcal{D}}}[\text{NN}_S^{(y)}(x) = L(x)] \geq 1 - \varepsilon \tag{13}$$

Now, we have

$$\{(\text{NN}_S^{(y)}(x), L(x))\}_{\substack{S\sim\mathcal{D}^n \\ x,y\sim\mathcal{D}}} \equiv \{(\widehat{y_i}, L(x))\}_{\substack{S\sim\mathcal{D}^n \\ \widehat{x_i},\widehat{y_i}\leftarrow\text{NN}_S(x) \\ x,y\sim\mathcal{D}}} \tag{14}$$

$$\approx_\varepsilon \{(\widehat{y_i}, L(\widehat{x_i}))\}_{\substack{S\sim\mathcal{D}^n \\ \widehat{x_i},\widehat{y_i}\leftarrow\text{NN}_S(x) \\ x,y\sim\mathcal{D}}} \tag{15}$$

$$\approx_\delta \{(\widehat{y_i}, L(\widehat{x_i}))\}_{\widehat{x_i},\widehat{y_i}\sim\mathcal{D}} \tag{16}$$

Line (15) is by distinguishability, since $\Pr[L(x) \neq L(\widehat{x_i})] \leq \varepsilon$. And Line (16) is by the regularity condition. $\qquad\square$

---

[5]We estimate the expectation in $M(x, y)$ by training an ensemble of 5000 pairs of classifiers $(f_1, f_2)$, each pair on disjoint train samples.

E.2   AGREEMENT PROPERTY

**Theorem 2** (Agreement Property). *For a given distribution $\mathcal{D}$ on $(x, y)$, and given number of train samples $n \in \mathbb{N}$, suppose NN satisfies the following regularity condition: If we sample two independent train sets $S_1, S_2$, then the following two "couplings" are statistically close:*

$$\{(x_i, \mathrm{NN}_{S_2}(x_i))\}_{\substack{S_1 \sim \mathcal{D}^n \\ S_2 \sim \mathcal{D}^n \\ x_i \in_R S_1}} \quad \approx_\delta \quad \{(\mathrm{NN}_{S_1}(x), \mathrm{NN}_{S_2}(x))\}_{\substack{S_1 \sim \mathcal{D}^n \\ S_2 \sim \mathcal{D}^n \\ x \sim \mathcal{D}}} \tag{17}$$

*The LHS is simply a random test point $x_i$, along with its nearest-neighbor in the train set. The RHS produces an $(x_i, x_j)$ by sampling two independent train sets, sampling a test point $x \sim D$, and producing the nearest-neighbor of $x$ in $S_1$ and $S_2$ respectively.*

*Then:*

$$\Pr_{\substack{S \sim \mathcal{D}^n \\ (x,y) \sim \mathcal{D}}} [\mathrm{NN}_S^{(y)}(x) = y] \approx_\delta \Pr_{\substack{S_1 \sim \mathcal{D}^n \\ S_2 \sim \mathcal{D}^n \\ (x,y) \sim \mathcal{D}}} [\mathrm{NN}_{S_1}^{(y)}(x) = \mathrm{NN}_{S_2}^{(y)}(x)] \tag{18}$$

*Proof.* Let the LHS of Equation (17) be denoted as distribution $P$ over $\mathcal{X} \times \mathcal{X}$. And let $Q$ be the RHS of Equation (17). Let $\mathcal{D}_x$ denote the marginal distribution on $x$ of $\mathcal{D}$, and let $p(y|x)$ denote the conditional distribution with respect to $\mathcal{D}$.

The proof follows by considering the sampling of train set $S$ in the following order: first, sample all the $x$-marginals: sample test point $x \sim \mathcal{D}_x$ and train points $S_x \sim \mathcal{D}_x^n$. Then compute the nearest-neighbors $\widehat{x} \leftarrow \mathrm{NN}_{S_x}(x)$. And finally, sample the *values* $y$ of all the points involved, according to the densities $p(y|x)$.

$$\Pr_{\substack{S \sim \mathcal{D}^n \\ (x,y) \sim \mathcal{D}}} [\text{NN}_S^{(y)}(x) = y] = \mathbb{E}_{\substack{S \sim \mathcal{D}^n \\ (x,y) \sim \mathcal{D}}} [\mathbb{1}\{\text{NN}_S^{(y)}(x) = y\}] \tag{19}$$

$$= \mathbb{E}_{\substack{S_x \sim \mathcal{D}_x^n \\ x \sim \mathcal{D}_x \\ \widehat{x} \leftarrow \text{NN}_{S_x}(x)}} \underbrace{\left[ \mathbb{E}_{\substack{y \sim p(y|x) \\ \widehat{y} \sim p(y|\widehat{x})}} [\mathbb{1}\{\widehat{y} = y\}] \right]}_{T(x,\widehat{x})} \tag{20}$$

$$= \mathbb{E}_{\substack{S_x \sim \mathcal{D}_x^n \\ x \sim \mathcal{D}_x \\ \widehat{x} \leftarrow \text{NN}_{S_x}(x)}} [T(x,\widehat{x})] \tag{21}$$

$$= \mathbb{E}_{(x_1,x_2) \sim P} T(x_1, x_2) \qquad (P: \text{LHS of Equation (17)})$$

$$\approx_\delta \mathbb{E}_{(x_1,x_2) \sim Q} T(x_1, x_2) \qquad (Q: \text{RHS of Equation (17)})$$

$$= \mathbb{E}_{\substack{S_1 \sim \mathcal{D}_x^n \\ S_2 \sim \mathcal{D}_x^n \\ x \sim \mathcal{D}_x \\ \widehat{x_1} \leftarrow \text{NN}_{S_1}(x) \\ \widehat{x_2} \leftarrow \text{NN}_{S_2}(x)}} [T(\widehat{x}_1, \widehat{x}_2)] \tag{22}$$

$$= \mathbb{E}_{\substack{S_1 \sim \mathcal{D}_x^n \\ S_2 \sim \mathcal{D}_x^n \\ x \sim \mathcal{D}_x \\ \widehat{x}_1 \leftarrow \text{NN}_{S_1}(x) \\ \widehat{x}_2 \leftarrow \text{NN}_{S_2}(x)}} \left[ \mathbb{E}_{\substack{\widehat{y_1} \sim p(y|\widehat{x}_1) \\ \widehat{y_2} \sim p(y|\widehat{x}_2)}} [\mathbb{1}\{\widehat{y_1} = \widehat{y_2}\}] \right] \tag{23}$$

$$= \mathbb{E}_{\substack{S_1 \sim \mathcal{D}^n \\ S_2 \sim \mathcal{D}^n \\ (x,y) \sim \mathcal{D}}} [\mathbb{1}\{\text{NN}_{S_1}^{(y)}(x) = \text{NN}_{S_2}^{(y)}(x)\}] \tag{24}$$

$$= \Pr_{\substack{S_1 \sim \mathcal{D}^n \\ S_2 \sim \mathcal{D}^n \\ (x,y) \sim \mathcal{D}}} [\text{NN}_{S_1}^{(y)}(x) = \text{NN}_{S_2}^{(y)}(x)] \tag{25}$$

as desired. $\qquad\qquad\qquad\qquad\qquad\qquad\qquad\qquad\qquad\qquad\qquad\qquad\qquad\qquad\square$

## F  NON-INTERPOLATING CLASSIFIERS: APPENDIX

Here we give an additional example of distributional generalization: in kernel SVM (as opposed to kernel regression, in the main text).

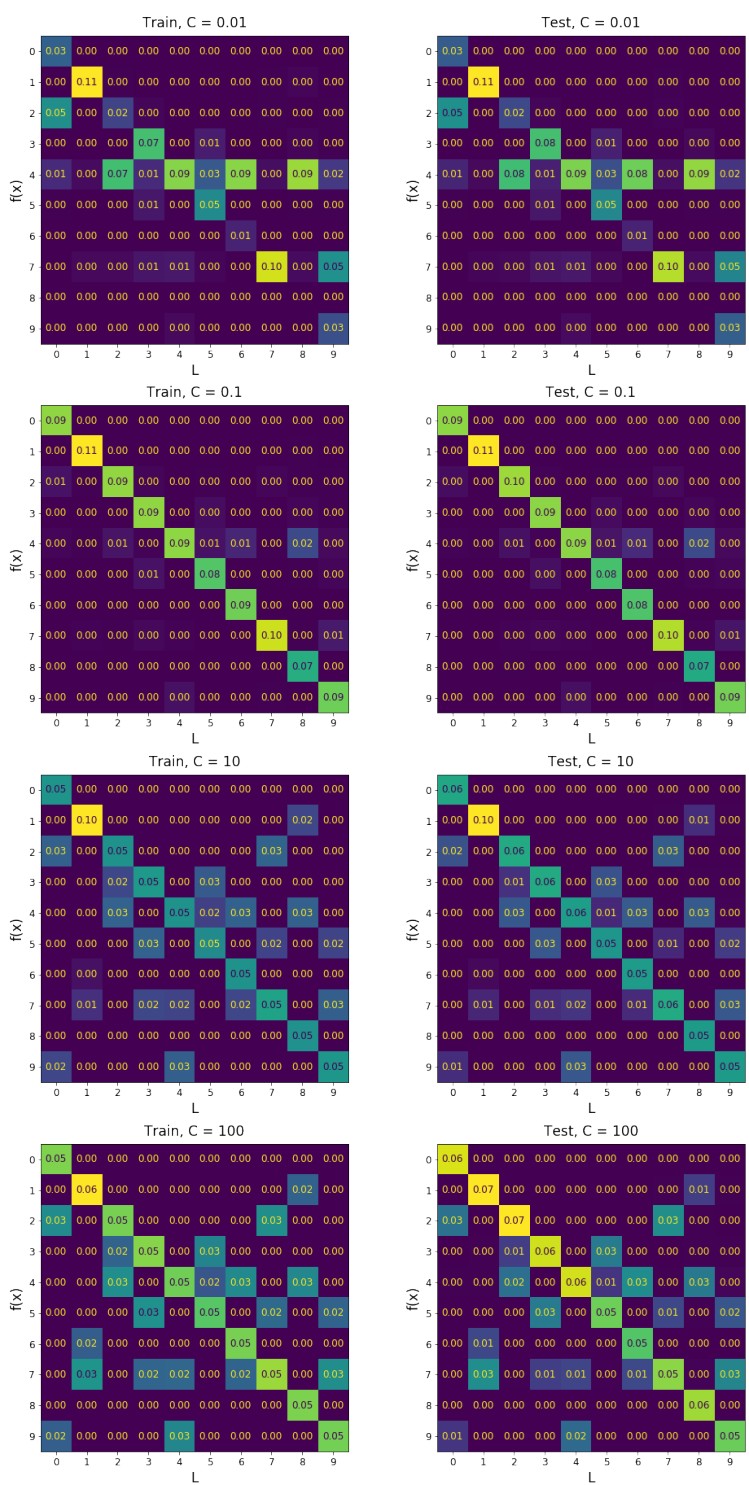

Figure 21: **Distributional Generalization.** Train (left) and test (right) confusion matrices for kernel SVM on MNIST with random sparse label noise. Each row corrosponds to one value of inverse-regularization parameter $C$. All rows are trained on the same (noisy) train set.

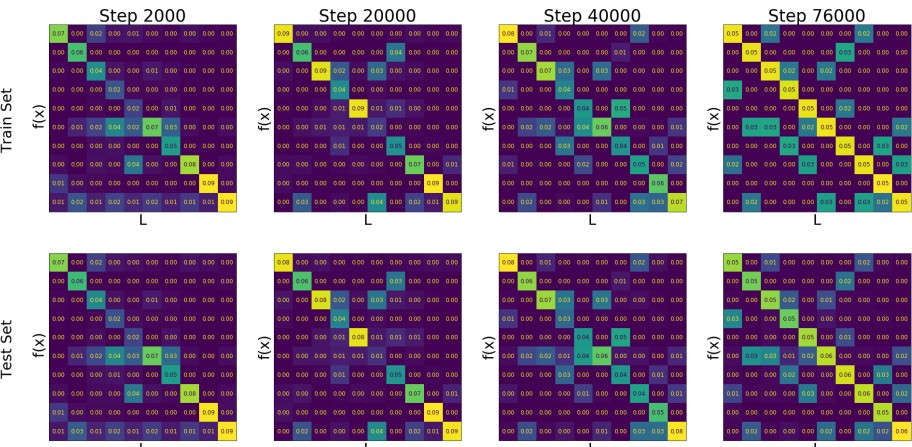

Figure 22: **Distributional Generalization for WideResNet on CIFAR-10.** We apply label noise from a random sparse confusion to the CIFAR-10 train set. We then train a single WideResNet28-10, and measure its predictions on the train and test sets over increasing train time (SGD steps). The top row shows the confusion matrix of predictions $f(x)$ vs true labels $L(x)$ on the train set, and the bottom row shows the corresponding confusion matrix on the test set. As the network is trained for longer, it fits more of the noise on the train set, and this behavior is mirrored almost identically on the test set.

