# OpenReview forum: "Distributional Generalization: A New Kind of Generalization"
_ICLR.cc/2021/Conference — Reject_

### Official Review · AnonReviewer3 · 2020-10-27

**Rating:** 7
**Confidence:** 3

**Review:**

This paper proposes an extended notion of generalization. The new proposed notion asks that for a family of tests $T: X \times Y → [0,1]$, $T(x, f(x))$  will be similar for train/test examples. The paper proposes three interesting conjectures that are related to distributional generalization. The paper proves Conjecture 1 for nearest-neighbor classifiers. The paper also gives empirical evidence supporting their conjectures.

I think this is a nice contribution. It raises some new questions on generalization, and points to some interesting properties that interpolating classifiers satisfy (at least empirically) that deserve to be studied in more detail. In particular, Conjecture 1 seems to suggest that there is a form of hierarchical learning that interpolating classifiers are doing without being told to explicitly do so.

---

> ### Author Response · Authors · 2020-11-12
> **Response to R3**
>
> Thank you for your review; we are glad you appreciate the empirical contribution of this work.
>
> We also agree that Conjecture 1 suggests some deeper connection between interpolating classifiers and hierarchal learners — we hope our work can inspire future study into these mechanisms.

---

### Official Review · AnonReviewer1 · 2020-10-28
**Review of Distributional Generalization**

**Rating:** 4
**Confidence:** 3

**Review:**

> Summarize what the paper claims to contribute

This paper generalizes the classical notion of generalization to the notion that the output of the classifier should be close when applied to the training data and the testing data. Two conjectures were made to predict the behavior of the distributional generalization for a few models and data. The first conjecture, Feature Calibration Conjecture, asserts that the distribution of the outputs are similar up to “distinguishable features”. The second conjecture, Agreement Conjecture, asserts that test accuracy matches the classification stability for interpolated classifiers. A number of experiments surrounding these conjectures were conducted to illustrate the Distributional Generalization

> List strong and weak points of the paper. Be as comprehensive as possible.
+ Great idea to look into distribution of the output beyond the error. I think this is long overdue and this paper has a lot of potential.
+ Comprehensive numerical studies which show some interesting findings.
- Many vague and ambiguous notations which make the paper hard to read.
- Barely any theoretical justification.
- "Too many" ideas. I would suggest the authors to divide the paper into 3 papers and dig deeper in each topic.

> Provide supporting arguments for your recommendation.

1. Notation. While the display on the bottom of page 2 is supposed to be an informal conjecture, it may be too informal. Does x∈TestSet indicate a distribution or a set? What does ≡ mean here? It was mentioned that the joint distribution of X and Y is D, and D^n denote n iid samples from D. But is D^n a sample of training data points, or a distribution? Later on it was mentioned that "S ∼ D^n" which does not make sense if D^n is a sample.
2. There are only two theoretical results in the paper with rigorous proofs. But they are established for nearest neighbor classifier only and some assumptions are imposed which warrants the results straightforwardly. However, most of the conjectures are made for many classifiers beyond nearest neighbor.
3. Indistinguishability and interpolated classifier: in the Indistinguishability meta-conjecture on page 2, the second equality is clearly due to interpolated classifier. I wonder if the closeness indicated in the first approximation (between trainset and testset) is a merit of the interpolated classifier, or the general good generalization performance of a classifier. It is not clear to me why interpolated classifiers will induce this closeness.
4. The role of L. From what I can see, this intermediate feature L is introduced because \mathcal{X} can be a large high-dimensional space which can make visualization or quantification of the closeness difficult. This function L is introduced so that it is easier to see how (L(x), f(x)) is distributed. For constant partition (L(x) = 0), the distributional generalization reduces to the marginal distribution of "f(x)". In my opinion, if a classifier is true generalizable, the distribution of (x, f(x)) itself, without L, should be the same between the test and training domains. From the statement of Theorem 1, it seems to me that  Conjecture 1 holds for nearest neighbor BECAUSE some distinguishable features exist, which may already be a strong assumption. So strong that the very existence of the distinguishable features warrants the generalizability. Unfortunately because there are only conjectures and no rigorous proofs, this part is still not clear to me.

> Ask questions you would like answered by the authors to help you clarify your understanding of the paper and provide the additional evidence you need to be confident in your assessment.

While the numerical studies have convinced me that distributional generalization is really a thing here, I am not sure that I am that impressed. After all, the precision was only two decimal points. Can the total-variation distance be calculated? Is there a counterexample in which one classifier does have classical generalizability but not distributional generalizability? Focus on a narrower set of topics but dig deeper may go a long way here.

> Minor comments

Page 3: “do we this same procedure” is a typo
Page 4: how is “learnable” defined?
Page 5: f←Train F (D^n) is not defined, esp. given that it is unclear if D^n is a distribution or a set. Should D^n be replaced by S?
Page 5: Indistinguishably is a typo

---

> ### Author Response · Authors · 2020-11-12
> **Response to R1**
>
> Thanks for your feedback. We believe there may have been some misunderstandings about crucial aspects of our results, which we clarify below.
>
>
> - Re. "The role of L”:  The notion of L is not defined just for experimental convenience — it is fundamental to the result. Formally, specifying a set of L is equivalent to specifying a notion of closeness between two probability distributions (known as an “Integral Probability Metric” in the literature). There are many cases where the train and test distributions are NOT close in total variation, as you suggest. But they can be close in other weaker metrics, which is captured by L. (See the “coarse partition” experiments in Section 3.1). For instance, we know that fully connected networks don’t generalize on images (so they will not be close in total variation distance). Our conjecture gives us a precise way to characterize how fully connected networks may be expected to generalize (for eg: they will still match the marginal probabilities as shown in Figure 7 or in the Animal-Object partition in Figure 14)
> Finally, you claim that the existence of distinguishable features is a strong assumption. This is not true, since the constant function L=0 is always a distinguishable feature (as we explain in Section 3.1).
> - Re “counterexample in which classifier has classical generalizability but not distributional generalizability”: This is trivially impossible — if the function generalizes perfectly then f(x)=y everywhere, which means that the joint distributions (x, y) and (x, f(x)) are identical. This implies that distributional generalization trivially holds (since for every function L, (L(x), y) and (L(x), f(x)) will also be identical)
> - Re. “It is not clear to me why interpolated classifiers will induce this closeness.”: The first approximate-equality in the Meta-conjecture is not specific to interpolating classifiers, but is simply the statement of distributional generalization. We explain this in the Introduction, and Section 5 shows how to apply this to non-interpolating models, which should answer your question.
> - Re. your question on total-variation distances: Conjecture 1 makes a quantitative prediction about the total-variation distance between coarsened train and test distributions (it is upper bounded by epsilon). Our experiments confirm that this closeness is within the predicted bounds. To further illustrate this, we will include an additional plot that shows the total variation distance as a function of the epsilon.
> - Re Notation: We use standard notation for sampling from probability distributions, but we will clarify the notation further in the next update. For clarification, $D^n$ denotes a product distribution over n samples from the distribution D. $S$ is a random variable over $(X \times Y)^n$ that is sampled from the distribution $D^n$. The notation $A \equiv B$ means distributions $A$ and $B$ are identical.
> - Re “Rigorous proofs”: Please see our common response regarding the notion of theoretical justification —  this is primarily an empirical paper.
>
> We are glad our work convinced you that “distributional generalization is really a thing.”
> We believe this is a significant contribution towards understanding interpolating classifiers.
> To help us understand your score, could you please clarify if your concerns are about the validity of the results, or their novelty?

---

### Official Review · AnonReviewer2 · 2020-10-28
**Interesting observations**

**Rating:** 6
**Confidence:** 3

**Review:**

This paper proposes a new notion of generalization called distributional generalization which states that the outputs of the classifier for train and test are close as distributions not just their corresponding accuracy numbers. They propose conjectures about their the distributional closeness that they expect and how it depends on the model, number of data points and the algorithm. This paper gives experiments to support their conjectures for different model classes including neural networks, kernel methods and decision trees.

I find the notion of distributional generalization interesting. The first experiment of this paper is that the label noise introduced in the training set in one subgroup of a particular class is also localized in the same subgroup in the test set although the classifier does not have any explicit information about the subgroups which is interesting. The paper formally states that the distribution of train and test outcomes are similar in distribution with respect to tests which themselves can learned by that model class with the current number of samples.

I recommend accepting this paper because the notion of distributional generalization that this paper introduces is both interesting and surprising. Moreover, this paper has shown that it holds widely across different interpolating classifiers like decision trees, neural networks and kernel methods.

The paper includes an extensive set of experiments that are easy to follow.

One concern is as itself mentioned in the paper that this work does not talk about what conditions on the distribution, algorithm and the model class are necessary for this conjecture to hold either empirically or theoretically and the conjecture is not precisely defined. Any insights on this part would be good to include.

Another concern is that although this is an interesting observation, the paper does not talk about why and how studying this form of generalization would be useful for understanding interpolating classifiers and generalization in general.

Questions:
1) The authors suspect locality to be the underlying reason behind distributional generalization and find this observation to be true for kernel methods also. Do the authors know if there is some work on the locality aspects of kernel methods arguing that some particular kernels are more local than others and how this relates to the observations in this paper?
2) Regarding the agreement property in section 4, I find it a little surprising that the conjecture says that the correlation and the accuracy are the same. Does this relate to the number of classes present?

---

> ### Author Response · Authors · 2020-11-12
> **Response to R2**
>
> Thank you for your feedback, and for recognizing that our results are “both interesting and surprising.”
>
> Regarding your main concerns:
>
> Re. “conditions on the distribution, algorithm, and model class”:
> This limitation is not unique to our work, but common to almost all works in the science of deep learning. That is, precisely stating these conditions amounts to formalizing exactly what is “real” about real-world distributions (as opposed to synthetic ones), and what is special about models and optimizers used in practice (as opposed to contrived ones).
> Some papers approach this by considering toy models of real distributions & architectures, such as gaussian inputs and random features. However, instead of passing to toy models, we prefer to deal only with real data and models throughout. This buys us realism, but comes at the necessary cost of formalism.
>
> Re. “why studying this form of generalization would be useful for understanding interpolating classifiers and generalization in general”:
>
> Several main reasons:
>
> 1. First, if we care about understanding interpolating classifiers, then the first step is to understand their empirical behaviors. That is, we should first understand what structural properties are true about these classifiers before we can prove them theoretically. Our empirical work characterizes classifier outputs in a more fine-grained way(*) than any previous work we are aware of.
> 2. Potential mechanism: As Reviewer 3 also suggests, our work indicates that interpolating classifiers implicitly learn to behave differently on ‘distinguishable features' even when they are not provided labels for these features explicitly. Mechanisms behind this type of empirical behavior may be useful in understanding the “implicit bias” of deep learning.
> 3. In theory, it can sometimes be easier to prove a stronger statement than a weaker one, because the stronger statement is “more true” and holds more universally. This is the hope behind our work: though Distributional Generalization (DG) is stronger than classical generalization, it appears to be more robust, and holds more universally. Thus, DG may be an easier object to reason about even if we care about classical generalization.
>
> (*): For example, as noted by R1, most prior works consider only a single scalar — Test Risk or Test Error — while we study the entire Test Distribution.
>
> We will include additional discussion about this in the next revision.
>
>
> Regarding your questions:
>
> 1. Regarding locality: The notion of locality is often used in kernels in the context of Nadaraya-Watson kernel smoothers (as discussed in the full related works in Appendix A). However, these are very different objects from what we consider in this paper, which are kernel regressors. We are not aware of prior work on the locality properties of kernel regressors.
> 2. Regarding the agreement property: We agree this is a very surprising property. It did not depend on the number of classes in our experiment. We actually attempted to uncover the mechanisms behind the agreement property in the course of this research — we did not succeed, but we refuted several potential mechanisms. We will include these investigations in an Appendix in the next revision, in case you are interested.

---

### Official Review · AnonReviewer4 · 2020-10-28
**Recommendation to Reject**

**Rating:** 5
**Confidence:** 3

**Review:**

This paper introduces a new notion of "distributional generalization" as a tool to quantify the difference between the outputs from training and testing data sets using a certain machine learning algorithm. The authors formulate two conjectures for the so-called "Interpolating classifiers": the Feature Calibration Conjecture and Agreement Conjecture. The conjectures are supported numerically by some real data experiments and are proven for the 1-nearest-neighbor classifier under some technical conditions.

Strengths:
1. Two conjectures are interesting and have the potential to explain some unexplained phenomena for some existing machine learning algorithms.
2. Numerical experiments are quite thorough, using different data sets and different machine learning algorithms.

Weaknesses:
1. The justifications of the conjectures are almost purely empirical (except in the 1-nearest-neighbor (1-N-N) classifier case). However, the 1-NN classifier is just a toy example that has limited use in machine learning literature. In fact, one even does not need to train the 1-NN classifier given a training data set. This is probably why it is easy to analyze 1-NN under the considered setting.
2. While observations from data experiments are interesting and I enjoyed reading them, the paper fails to directly demonstrate how these observations can be used to improve performances or our understandings of existing algorithms. For example, how Conjecture 2 is useful? can we use it to evaluate the uncertainty (or confidence interval) of the test accuracy?
How can we find distinguishable features in Conjecture 1 in practice? using trees?

Conclusion:
Although I find the proposed conjectures interesting and may have potential values, the current formulation, and justification for these conjectures are a bit too superficial and loose. It is difficult for me to envision a scenario where these conjectures can make a meaningful impact.

---

> ### Author Response · Authors · 2020-11-12
> **Response to R4**
>
> Thank you for recognizing that our conjectures are interesting, and our experiments thorough.
> Regarding the lack of theoretical justification, please see our common response to all reviewers, which clarifies that this paper should be seen as primarily an empirical work.
>
> We address your other concerns below:
>
> Regarding the practical usefulness: The primary objective of this paper is scientific — to help understand the behavior of methods used in practice. We do not aim to directly improve current methods, but rather to provide a lens for studying and understanding them.
> This is in the same vein as papers such as [Zhang et al], which have led to further theoretical and empirical insights into generalization in deep learning.
>
> For Conjecture 2 specifically: We did investigate whether this property can be used for calibrated uncertainty estimates, but unfortunately this does not work for very interesting and subtle reasons (briefly, Conjecture 2 holds “on average” over inputs, but not “pointwise”, which is what you need for pointwise uncertainty estimates). We omitted these experiments from the submitted version, but we will update the Appendix to include these investigations.
>
> For Conjecture 1:
> We agree it would be valuable to enumerate the set of all distinguishable features. This may be infeasible, because this set could be very large (even exponentially large in natural parameters).
> Instead, we give a procedure for “testing” if a given partition L is a distinguishable feature (Definition 1). So, for any candidate feature, you can efficiently test if it is distinguishable or not.
>
> Regarding the claim that the current formulation of the conjecture is “too loose”:
> Note that Conjecture 1 makes a precise, quantitative claim about the total-variation distance between two distributions, in terms of testable problem parameters (the epsilon-distinguishability of the feature).  This Conjecture captures the correct quantitative behavior for all of the experiments in Section 3.1. Each of the experiments in 3.1 highlights a different aspect of interpolating classifiers, and our Conjecture correctly predicts all of these behaviors.
>
> We hope that given your positive impression of our conjectures and experiments, the score can be increased to ‘accept’ if you are satisfied with our answers to your questions.
>
>
>
>
> [Zhang et al.] “Understanding deep learning requires rethinking generalization”.  ICLR 2017.

---

### Author Response · Authors · 2020-11-12
**Common Response to All Reviewers**

We thank all the reviewers for their feedback. We clarify some common criticisms raised by several reviewers:

1. "Theoretical justification": Our paper should be seen as primarily an empirical work, which stands as an independent contribution even without a theoretical proof for the conjectures. Our motivation for stating the formal conjecture is to precisely characterize the experimental observations, and to make future testable predictions. We hope this conjecture can eventually be proven, but that is not our objective in this work. (It is extremely rare in deep learning to have fully rigorous proofs about realistic experimental setups)
    There have been numerous examples of works in deep learning like [Zhang et al.], [Nakkiran et al.] where extensive experiments were used to identify interesting behavior that should be further understood theoretically.

2. “Usefulness in understanding existing algorithms”: Our paper advances our understanding simply because it teaches us something new about existing algorithms — something previously unknown, and fairly universal. Moreover, our work suggests deeper mechanisms at play in interpolating classifiers, which we don’t yet understand. For example, as Reviewer 3 points out, interpolating classifiers implicitly learn to behave differently on “distinguishable features” even when they are not provided labels for these features explicitly. While understanding the exact mechanisms is beyond the scope of our paper, this empirical behavior is one characterization of the “implicit bias” of deep learning.

[Zhang et al.] “Understanding deep learning requires rethinking generalization”.  ICLR 2017.

[Nakkiran et al.] “Deep Double Descent: Where Bigger Models and More Data Hurt”.  ICLR 2020.

---

> ### Author Response · Authors · 2020-11-23
> **Revised PDF uploaded**
>
> We have updated the PDF with the following additions, following reviewer comments:
>
> - Added Appendix C.7 and Figure 15, to illustrate the concrete quantitative prediction of Conjecture 1. We show experimentally that the TV-distance between the joint distributions $(L(x), f(x))$ and $(L(x), y)$ is at most $\epsilon$, as predicted by our conjecture. We hope this addresses concerns of Reviewer 1 and Reviewer 4 that the conjecture is only qualitative.
> - Added Appendix D.3, which explores (and eventually refutes) two potential mechanisms which could explain the “Agreement Property”. This shows the subtlety of this property, and helps support future work into the mechanisms behind it.
> - We expanded the Conclusion to discuss how our work relates to the study of Classical Generalization. We also discuss how our results should update certain folklore intuitions about “memorization” in interpolating models.
> - We added a brief list of open questions raised by our work (Section 6.1)

---

### Decision · Program_Chairs · 2021-01-07
**Final Decision**

**Decision:**

Reject

**Comment:**

The paper introduces a notion of distributional generalization, which aims at characterizing aspects of underlying distribution that are learned by a trained predictor. Authors make several interesting conjectures and support them with empirical evidence. Reviewers agreed on the novelty of the ideas; however, the work seems to be preliminary in its current form. Unfortunately, I cannot recommend acceptance at this time.